# Neural mechanisms of learned suppression uncovered by probing the hidden attentional priority map

**Changrun Huang[1,2]\*, Dirk van Moorselaar[1,2], Joshua Foster[3], Mieke Donk[1,2], Jan Theeuwes[1,2,4]**

[1]Department of Experimental and Applied Psychology, Vrije Universiteit Amsterdam, Amsterdam, Netherlands; [2]Institute Brain and Behavior, Amsterdam, Netherlands; [3]Department of Psychological and Brain Sciences, Boston University, Boston, United States; [4]William James Center for Research, ISPA – Instituto Universitario, Lisbon, Portugal

**eLife Assessment**

This **important** study uses recently developed EEG analysis methods to investigate spatial distractor suppression in a combined visual search/working memory task. The reported results are **compelling**, although they are open to multiple interpretations. The study will be of interest to cognitive neuroscientists and psychologists working on visual attention and memory.

**Abstract** Attentional capture by an irrelevant salient distractor is attenuated when the distractor appears more frequently in one location, suggesting learned suppression of that location. However, it remains unclear whether suppression is proactive (before attention is directed) or reactive (after attention is allocated). Here, we investigated this using a 'pinging' technique to probe the attentional distribution before search onset. In an EEG experiment, participants searched for a shape singleton while ignoring a color singleton distractor at a high-probability location. To reveal the hidden attentional priority map, participants also performed a continuous recall spatial memory task, with a neutral placeholder display presented before search onset. Behaviorally, search was more efficient when the distractor appeared at the high-probability location. Inverted encoding analysis of EEG data showed tuning profiles that decayed during memory maintenance but were revived by the placeholder display. Notably, tuning was most pronounced at the to-be-suppressed location, suggesting initial spatial selection followed by suppression. These findings suggest that learned distractor suppression is a reactive process, providing new insights into learned spatial distractor suppression mechanisms.

**\*For correspondence:**
changrunhuang@gmail.com

**Competing interest:** The authors declare that no competing interests exist.

## Introduction

Even though large amounts of information constantly bombard our senses, we can effortlessly direct attention to relevant information and ignore information that may distract us. Recent studies have demonstrated that attentional selection can be so efficient because visual input is highly repetitive and structured, which makes it possible to predict what information will appear next based on the current sensory input (*Friston, 2009*; *Kok et al., 2017*). Extracting these regularities from the environment, often called statistical learning (*Chun and Jiang, 1999*; *Frost et al., 2015*), occurs effortlessly across trials, operates largely outside the realm of conscious awareness, and is not contingent on explicit knowledge of the regularity (*Gao and Theeuwes, 2022*; *Goujon et al., 2015*; *Turk-Browne*

*et al., 2005*; but see *Vicente-Conesa et al., 2023*). In addition to top-down and bottom-up control processes (*Corbetta and Shulman, 2002*; *Desimone and Duncan, 1995*; *Theeuwes, 2010*), statistical learning plays a crucial role in attentional selection (*Awh et al., 2012*; *Failing and Theeuwes, 2018*; *Theeuwes et al., 2022*). According to the tripartite model of attention (*Awh et al., 2012*; *Theeuwes, 2019*; *Theeuwes et al., 2022*; *Theeuwes, 2025*), the interaction between top-down, bottom-up, and selection history, a category which encompasses statistical learning as well as other history-based effects, jointly determines the weights in a spatial priority map, which determines, in a winner-take-all fashion, which object is selected (*Chelazzi et al., 2019*; *Zelinsky and Bisley, 2015*). The present study investigates the mechanisms underlying statistical learning, specifically learned distractor suppression, which represents one critical subcomponent of selection history. While theoretical models like the tripartite framework and the recent monolithic theory (*Anderson, 2024*) offer complementary perspectives on attentional control, our investigation focuses on empirically characterizing the statistical learning mechanisms underlying learned distractor suppression.

Previous studies have demonstrated that observers can learn which location is most likely to contain the target. *Geng and Behrmann, 2005*, showed that targets presented in high-probability locations (HPL) are detected faster than those in low-probability locations (LPL) (see also *Ferrante et al., 2018*; *Jiang et al., 2013*). In a related vein, *Huang et al., 2022*, used the additional singleton visual search task in which the target was presented more often in one location than in all other locations. This task was interleaved with probe trials, enabling an exploration of the distribution of attention across the display in the period preceding search display onset. Crucially, the probe task showed spatial enhancement for the location with the highest likelihood of containing the target. Based on these findings, it was argued that the amplification of weights within the spatial priority map, driven by statistical learning, takes place preemptively, before the actual display presentation, implying that priority was changed prior to the allocation of attention. Consistent with this, an innovative EEG study by *Duncan et al., 2023*, employed a 'ping' technique, commonly used in the realm of working memory (*Wolff et al., 2015*; *Wolff et al., 2017*), to unveil the weights of the concealed attentional priority. This technique pushes a wave of activity, often via presentation of a high-contrast stimulus, through the visual system to reveal hidden neural representations within networks of altered synaptic weights. *Duncan et al., 2023*, demonstrated that, following the acquisition of a learned priority for a specific spatial location, there was reliable decoding of the high-probability target location in the evoked EEG signal when a visual ping occurred during the interval preceding the presentation of the search display. The above chance decoding is assumed to reveal the prioritized (enhanced) location within the 'activity-silent' priority map.

These previous studies indicate that people easily pick up and learn the statistical regularities associated with the location of the target. Recently, however, a large number of studies demonstrated that not only target but also distractor-based regularities affect attentional deployment. Employing a modified version of the additional singleton paradigm, where the color singleton distractor appeared with a higher probability at a specific location relative to other locations, Wang and Theeuwes demonstrated that distractor interference was reduced when distractors appeared at this high-probability distractor location as compared to other locations. Furthermore, participants exhibited slower responses when the target was presented at this high-probability distractor location (*Wang and Theeuwes, 2018b*; *Wang and Theeuwes, 2018a*; *Wang and Theeuwes, 2018c* also see *Ferrante et al., 2018*; *Goschy et al., 2014*), suggesting that the spatial distractor imbalance resulted in learned suppression of the high-probability distractor location (*Failing and Theeuwes, 2020*; *Feldmann-Wüstefeld et al., 2021*; *Ferrante et al., 2018*; *Goschy et al., 2014*; *Huang et al., 2021a*; *Sauter et al., 2021*; *Sauter et al., 2021*; *Allenmark et al., 2022*; *Zhang et al., 2022*; *van Moorselaar and Theeuwes, 2021b*; *van Moorselaar and Theeuwes, 2022*; *Wang and Theeuwes, 2018b*; *Wang and Theeuwes, 2018a*). Such learned distractor suppression is not restricted to the spatial domain, as many studies have demonstrated that distractor interference can be attenuated, or even completely eliminated, through learning about probable distractor features (*Gaspelin and Luck, 2018a*; *Stilwell and Gaspelin, 2021*; *Stilwell et al., 2022*; *Vatterott et al., 2018*; *Failing et al., 2019*; *Vatterott and Vecera, 2012*).

Despite the widespread acknowledgment that statistical learning of distractor characteristics can mitigate their interfering effects, the precise mechanisms and temporal dynamics of this process remain subjects of ongoing debate. As articulated by *Liesefeld et al., 2024*, suppression could potentially occur at three distinct time points: first, before the distractor appears, reflecting anticipatory

preparation; second, after distractor onset but before attentional capture, indicating rapid early filtering; or third, after attentional capture has occurred. A key point of contention in current research revolves around whether the suppression mechanism operates proactively (before attention is deployed, encompassing the first two scenarios) or reactively (after attentional capture, aligning with the third scenario).

To examine when suppression was instantiated, similar to the method used for target learning, *Huang et al., 2022*; *Huang et al., 2023*, randomly interleaved probe trials to explore whether learned distractor suppression was implemented before search display onset. Across multiple studies it was found that responses were slowed when the probe appeared at the high-probability distractor location (*Huang et al., 2022*; *Huang et al., 2021b*; *Kong et al., 2020*), even when the probe display was presented prior to the expected onset of the search display (*Huang et al., 2023*), leading to the conclusion that suppression operates prior to search display onset, and, thus, before the first shift of attention. However, others have argued that suppression can only be instantiated *reactively*, after attention has been directed to that location (*Chang et al., 2023*; *Moher and Egeth, 2012*). In this respect, it is noteworthy that in the capture-probes studies (*Huang et al., 2022*; *Huang et al., 2023*), probes that revealed suppression were presented after presenting a neutral placeholder display. Therefore, it is possible that while suppression may have been in place at the moment in time the additional singleton search display was presented, the earlier presented placeholder display may have triggered an initial shift of attention to the high-probability distractor location which was immediately followed by rapid attentional disengagement and suppression (*Theeuwes et al., 2000*; *Theeuwes and Chen, 2005*). Taking this concern into account, *Chang et al., 2023*, introduced a modified version of the capture-probe paradigm to examine the timing of suppression. They combined a learned spatial suppression search task with a randomly interleaved probe task. Unlike *Huang et al., 2022*; *Huang et al., 2023*, who employed an offset probe detection following a placeholder, *Chang et al., 2023*, used an onset bar discrimination task that eliminated the need for a placeholder. In this task, participants discriminated the orientation of a briefly presented tilted line, which was equally likely to appear at any of the search locations. Contrary to a proactive account, *Chang et al., 2023*, found that while suppression occurred during the search task, as indicated by reaction times (RTs), probe accuracy was higher at the high-probability distractor location compared to the LPL (see Experiment 2 in *Chang et al., 2023*). This suggests that the HPL was initially selected for attentional processing before suppression occurred, supporting a reactive suppression mechanism.

The current study was designed to establish how suppression is implemented within the attentional priority map. For this purpose, analogous to the pinging technique previously employed by *Duncan et al., 2023*, search display onset was preceded by a task-irrelevant neutral placeholder display that served as a visual ping (see *Figure 1*). The attentional profile elicited by this visual ping was reconstructed in a time-resolved manner with a spatial inverted encoding model (IEM) (*Brouwer*

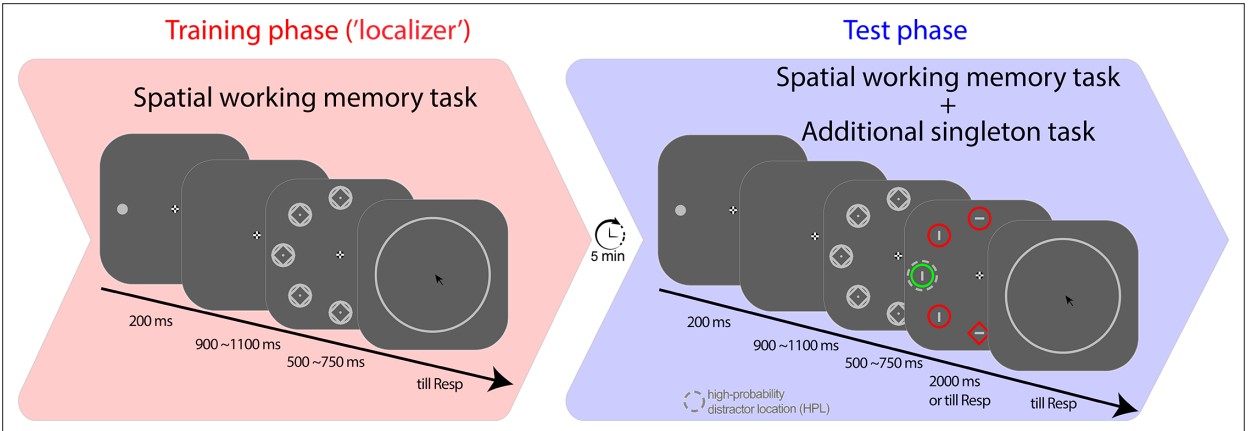

**Figure 1.** Schematic trial and experimental design. The experiment commenced with a training phase, during which participants were tasked with retaining the spatial cue's location in their memory for subsequent testing following a delay. After a 5 min break, the training phase transitioned to the test phase, where a search task was embedded during memory maintenance. Upon search display onset, participants were instructed to search for a unique shape singleton while ignoring the color singleton. Unbeknownst to the participants, the color singleton was presented more frequently at a location referred to as the high-probability location (HPL).

*and Heeger, 2009*; *Brouwer and Heeger, 2011*; *Foster et al., 2016*; *Foster et al., 2017*; *Sprague et al., 2016*), generating location-selective channel tuning functions (CTFs) over time. In order to model the spatial response profile associated with learned suppression, a prerequisite is that the high-probability distractor location periodically shifts across space such that the topographic distribution of different spatial channels (i.e. neural populations) can be considered. Periodically shifting the high-probability distractor location, however, inevitably introduces temporal confounds (*Duncan et al., 2023*), such as lingering and enduring effects from initial learning experiences on subsequent attentional biases (*Wang and Theeuwes, 2020*). In light of these challenges, we opted to not center the tuning profiles around the high-probability distractor location. Instead, the HPL remained static throughout the experiment. We then measured its influence on the priority landscape via a spatially specific modulation of a top-down attention signal, which can be flexibly adjusted on a trial-by-trial basis (*Posner et al., 1980*; *Theeuwes, 2019*).

To this end, we embedded the additional singleton task, which was always preceded by a placeholder display, within the maintenance period of a spatial working memory task (see *Figure 1*), with the spatial memory serving as a proxy for top-down attention (*Awh and Jonides, 2001*). Participants performed a visual search task wherein they were tasked with identifying a shape singleton in the presence of an irrelevant color singleton. Compared to all other locations, this color singleton appeared more frequently at a specific location, which was termed the HPL. Prior to the search task, we introduced a continuous recall spatial memory task to reveal the hidden attentional priority map. Participants had to memorize the location of a memory cue continuously and report this location after the visual search task. Critically, after the presentation of the memory cue but before the onset of the search display, a neutral placeholder display was presented to probe how the hidden priority map is reconfigured by the learned distractor suppression. This design not only enabled us to investigate whether a ping could unveil learned attentional biases associated with suppression but, crucially, also facilitated the examination of the tuning profile of this suppression in a time-resolved manner. This approach allows for a dissociation between proactive and reactive mechanisms. Specifically, within a proactive account, memory-specific tuning should be attenuated at the HPL immediately following placeholder onset (*Huang et al., 2022*; *Huang et al., 2021a*; *Kong et al., 2020*), whereas a reactive suppression account predicts that the to-be-suppressed location is initially attended resulting in temporarily enhanced tuning at that location (*Chang et al., 2023*; *Moher and Egeth, 2012*).

## Results

### Behavior: spatial distractor learning and precise spatial memory maintenance

We conducted a pairwise t-test comparing memory recall deviations between training and test phases to examine how memory performance varied with or without a concurrent secondary search task. Our findings revealed a significantly larger recall deviation at the test phase compared to the training phase (6.27°±0.95 vs 11.94°±3.55, $t(23)$ = 5.67, p<0.001, $d$=2.52), indicating that the process of tracking the spatial memory cue was substantially disrupted by the concurrent secondary task in the test phase. Note, that despite being disrupted by the search task, overall memory recall performance in the test phase was nevertheless high, indicating that observers were able to maintain a relatively precise memory representation outside the current focus of attention.

Next, we examined whether participants learned to inhibit the HPL in the visual search task while simultaneously maintaining an online representation of a spatial location in WM. A paired t-test was performed on mean RTs and accuracy, comparing trials where the distractor appeared at the HPL with those in which it appeared at one of the LPL [When contrasting RTs between trials featuring distractors at the HPL and those at the LPL, potential RT slowdown in the latter scenario could arise from the target occupying the HPL, a condition known to yield impaired target processing. To disentangle the effects of distractor suppression and target impairment and mitigate confounding, we categorized trials as low-probability only when both the target and distractor were not presented at the HPL] (see *Figure 2A and B*). The planned paired t-test revealed that participants exhibited faster ($t(23)$ = 10.14, p<0.001, $d$=0.43) and more accurate ($t(23)$ = 5.19, p<0.001, $d$=0.71) responses when the distractor appeared at the HPL ($M_{RT}$ = 965 ms, $M_{ACC}$ = 92.2%) compared to the LPL ($M_{RT}$ = 1017 ms, $M_{ACC}$ = 89.5%), showing a clear effect of statistically learned distractor suppression and replicating previous

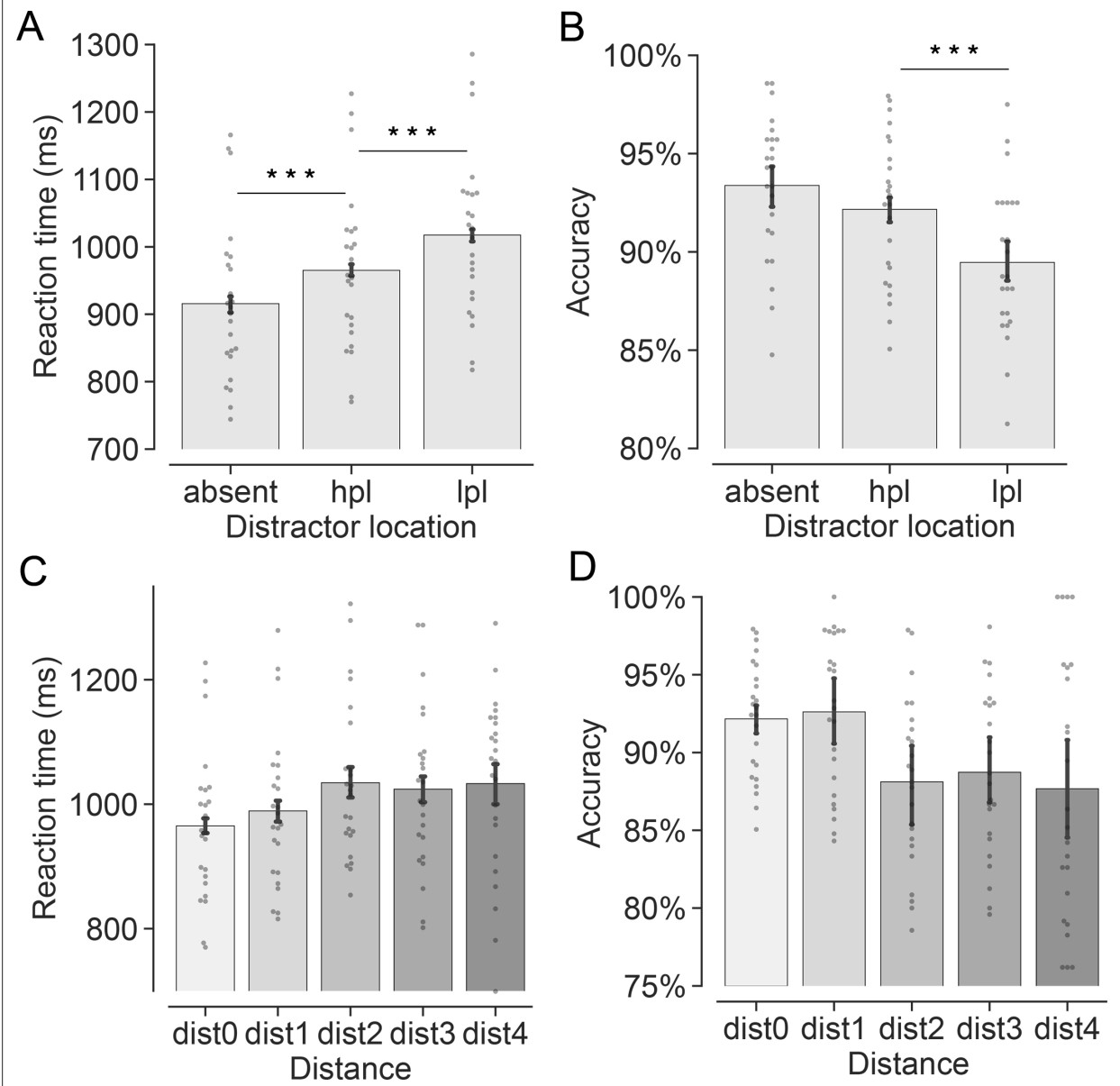

**Figure 2.** Behavioral findings in the search task. The top panels show (**A**) the mean reaction times (RTs) and (**B**) the mean accuracy under conditions where the distractor was absent, presented at either the high-probability location (hpl) or the low-probability locations (lpl). The bottom panels show (**C**) the mean RTs and (**D**) the mean accuracy in relation to the relative distance between the distractor and the high-probability location. Specifically, dist0, dist1, dist2, dist3, and dist4 signify instances where the distractor was at the hpl, one position, two positions, three positions, and four positions away from the hpl. Small gray dots show data for individual participants. The presence of '***' denotes statistical significance at the level of p<0.001. Error bars (N = 24) indicate condition-specific, within-subject 95% confidence intervals (**Morey, 2008**).

The online version of this article includes the following figure supplement(s) for figure 2:

**Figure supplement 1.** No evidence of behavioral interaction between spatial working memory and visual search.

findings (**Ferrante et al., 2018**; **Wang and Theeuwes, 2018b**). Further confirming the spatial component of suppression, a planned paired t-test on RTs for target location (collapsed across distractor-absent and distractor-present trials) demonstrated a substantial delay in participants' responses when the target was presented at the HPL ($M$=1018 ms), as opposed to the LPL ($M$=958 ms, $t(23)$ = 5.45, p<0.001, $d$=0.48). The same paired t-test on the mean accuracy ($M_{HPL}$ vs $M_{LPL}$: 90.1% vs 91.7%) did not reveal any difference ($t(23)$ = 1.68, p=0.106, $d$=0.36, $BF_{01}$=1.37).

To further characterize whether the learned suppression scaled with the relative distance to the HPL, we assigned the low-probability trials into one of four distance groups depending on the relative distance between the distractor and the HPL. The mean RTs and mean accuracy were submitted to a repeated measures ANOVA with a within-subject factor Distance (dist-1, dist-2, dist-3, dist-4). A significant main effect was found in the analysis of RTs ($F$(2.22, 51.06)= 3.87, p = 0.023, $\eta_p^2$ = 0.14). As shown in *Figure 2C*, responses in the dist-1 condition ($M_{dist\_1}$=989 ms) were faster than those in the dist-2 condition ($M_{dist\_2}$=1034 ms, $t$(23) = 3.96, p=0.003, $d$=0.36), and marginally faster than those in the dist-3 condition ($M_{dist\_3}$=1024 ms, $t$(23) = 2.83, p=0.058, $d$=0.28) and dist-4 condition ($M_{dist\_4}$=1033 ms, $t$(23) = 2.75, p=0.069, $d$=0.33). All other comparisons did not reach significance (all ps = 1). Mean accuracy analyses mimicked these findings (see *Figure 2D*), showing a main effect of Distance ($F$(2.13, 49)= 4.20, p = 0.019, $\eta_p^2$ = 0.15). Performance in the dist-1 condition ($M_{dist\_1}$=92.6%) was more accurate than in the dist-2 condition ($M_{dist\_2}$=88.1%, $t$(23) = 4.24, p=0.002, $d$=0.86), and in the dist-3 condition ($M_{dist\_3}$=88.7%, $t$(23) = 2.97, p=0.041, $d$=0.75), and marginally more accurate than in the dist-4 condition ($M_{dist\_4}$=87.7%, $t$(23) = 2.87, p=0.052, $d$=0.73).

Given that the current design inherently results in the distractor being more likely to repeat at the HPL than at LPL across consecutive trials, we assessed how this co-occurrence might influence our suppression findings. Consistent with previous research, we found that trials with repeated distractor locations were 41 ms faster ($t$(23) = 6.32, p<0.001, $d$=0.33) and 2.1% more accurate ($t$(23) = 4.21, p<0.001, $d$=0.86) compared to trials with non-repeated distractor locations. Importantly, excluding trials with repeated distractor locations did not alter the main findings related to distractor regularity or the distance effect, suggesting that intertrial effects alone cannot fully account for the learned suppression effects induced by spatial distractor imbalances.

In summary, observers remained sensitive to the spatial distractor regularity when the visual search task was embedded within the maintenance period of a spatial memory task. Consistent with previous work, distractor learning was characterized by a spatial gradient centered at the high-probability distractor location, where distractor interference was attenuated, and target processing was impaired. After having validated that processing at the high-probability distractor location was suppressed, we set out next to establish whether this suppression could be revealed in anticipation of search display onset, and critically, whether it did attenuate or enhance memory-specific tuning at the high-probability distractor location relative to the other display locations.

## Localizer data: alpha-band tuned pinging of spatial working memory representations

We utilized an IEM to reconstruct the attentional tuning profiles that track the spatial locations of the memory cue (see Methods for details). The resulting CTF profiles were estimated by their slopes, which index spatial selectivity, where higher CTF slopes indicate greater location selectivity. Given the robust association with spatial attention and working memory maintenance, these analyses focused on oscillatory power within the alpha-band (8–12 Hz).

Before investigating whether attentional tuning profiles elicited by the placeholder displays would be modulated by distractor-based learning, we first characterized the spatial memory code in the training phase (i.e. in the absence of selection history effects). If participants indeed maintained such spatial information in their working memory, pronounced CTF profiles should be elicited by the memory cue. Consistent with previous work (*Foster et al., 2016*; *Foster et al., 2017*; *van Moorselaar et al., 2018*), both evoked and total alpha power generated reliable CTFs in response to memory display onset (p<0.05), with evoked power returning to baseline relatively early in the delayed period (*Figure 3—figure supplement 1*), whereas total alpha power enabled reliable CTF reconstruction throughout the entire delay period (*Figure 3A*). Critically, as depicted in *Figure 3B*, the evoked CTFs were distinctly revived upon the presentation of the placeholder display (p<0.05).

Given that the primary objective of this study was to monitor location-specific modulations of the attention set as a function of the distance from the HPL, we also ensured that the observed reconstructions were not driven by a subset of locations, but instead were homogeneous across all possible locations. As visualized in *Figure 3C/D*, the generated CTFs were not the result of imbalanced spatial selectivity of specific locations, as similar tuning was observed across all eight individual locations. Indeed, an artificially created distance-based gradient centered around the location that would become the HPL in the subsequent test phase did not exhibit any unbalanced spatial selectivity

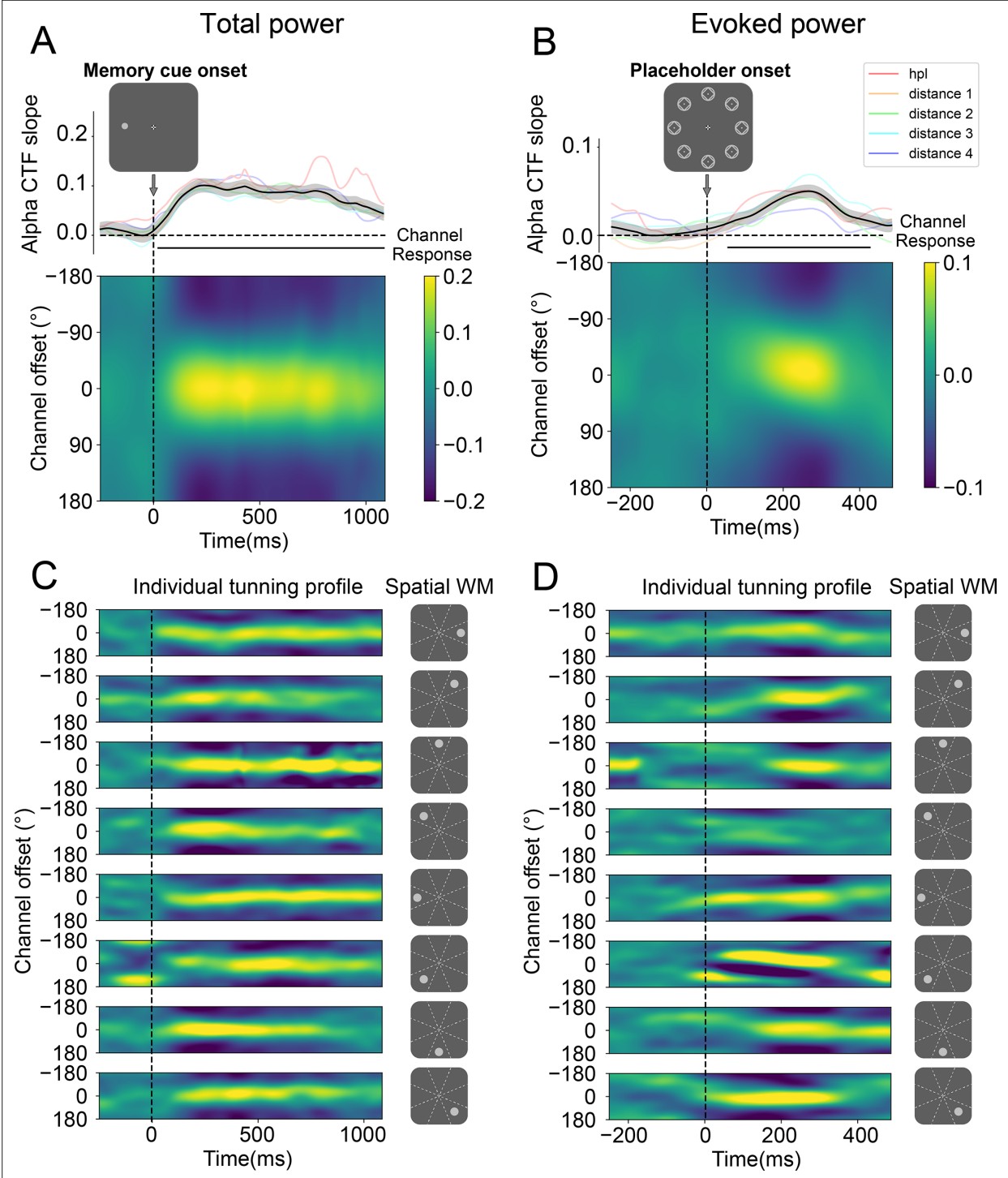

**Figure 3.** The performance of the localizer in the training phase. In the training phase, the average alpha-band channel tuning function (CTF) profiles reliably tracked the location of the memory cue following its onset (**A**). Although the spatial memory representation returned to baseline during the maintenance period, it was revived by a neutral placeholder display (**B**) that was irrelevant to the memory task. The lower images in (**A** and **B**) depict the responses across channels, while the plots above show CTF slopes, with amplitude indicating spatial selectivity. The light-colored lines in the background represent CTF slopes tuned to different memory cue locations, grouped by their distance from a location that will be the high-probability location in the test phase. Visual inspection reveals no spatial bias. Shaded areas represent bootstrapped standard error of the mean (SEM, N = 24). Time points showing significant differences in CTF slopes, identified through a cluster-based permutation test (p<0.05), are marked with horizontal black insets. (**C**) illustrates that individual CTF profiles, following the memory cue onset, were finely tuned to each of the eight spatial cue locations with

*Figure 3 continued on next page*

*Figure 3 continued*

comparable response strengths. (**D**) shows that individual CTF profiles tuned to each of the eight spatial cue locations were revived by the placeholder to a similar extent, although the effect was somewhat obscured in the fourth row of the panel.

The online version of this article includes the following figure supplement(s) for figure 3:

**Figure supplement 1.** The performance of the localizer in the training phase.

(see colored coded lines in *Figure 3A and B*). These analyses confirm that the data from the training set did not exhibit a spatial bias from the outset and can thus serve as a neutral independent dataset to investigate how distractor learning in the test phase would, if at all, modulate attentional tuning across all display locations.

## Cross-session encoding: no evidence for proactive suppression of top-down attentional selection biases

After having established that the model obtained from the training phase reliably tracked the spatial content of working memory across all eight locations without a spatial bias, we next examined how this model generalized to the test phase, where selection history effects were introduced via the embedded search task during memory maintenance. In doing so, we ensured that data points in the training and test sets were aligned such that training and testing were conducted on matching time samples. We first depicted the alpha CTFs from the onset of the memory cue to examine how it was encoded in the test phase, where the secondary search task was embedded. As depicted in *Figure 4*, the averaged tuning profiles resulting from this cross-session encoding procedure mimicked the pattern of the localizer, indicating that the training data could be used to successfully track the memoranda during the test phase with an intermediate search task. One main difference was observed on total power, such that it no longer tracked the memorized position throughout the entire delay interval but

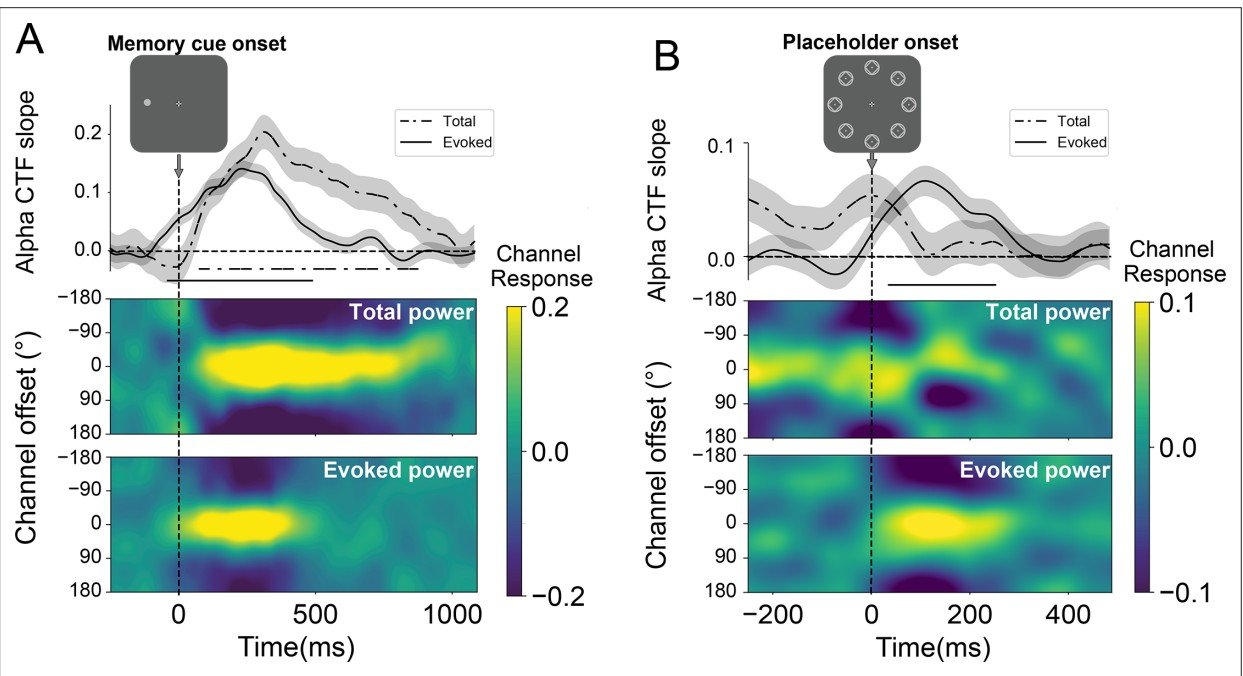

**Figure 4.** Cross-session encoding: alpha channel tuning functions (CTFs) of memory cue display and placeholder display. The location CTFs obtained by generalizing the trained model to the test set, ensuring strict alignment of time samples between the training and test phases. CTFs were reconstructed from evoked and total alpha power following the onset of (**A**) the memory cue display and (**B**) the placeholder display. The spatial location of the memory cue was reliably tracked by the alpha CTFs, which returned to baseline before the presentation of the placeholder display. The spatial information maintained in working memory was then revived upon the presentation of the placeholder display, selectively for evoked power. The lower image depicts the responses across channels, while the plot above shows CTF slopes, with amplitude signifying spatial selectivity. Shaded areas reflect bootstrapped SEM (N = 24). Time points exhibiting significant differences in CTF slopes, identified through a cluster-based permutation test (p<0.05), are marked with horizontal black insets.

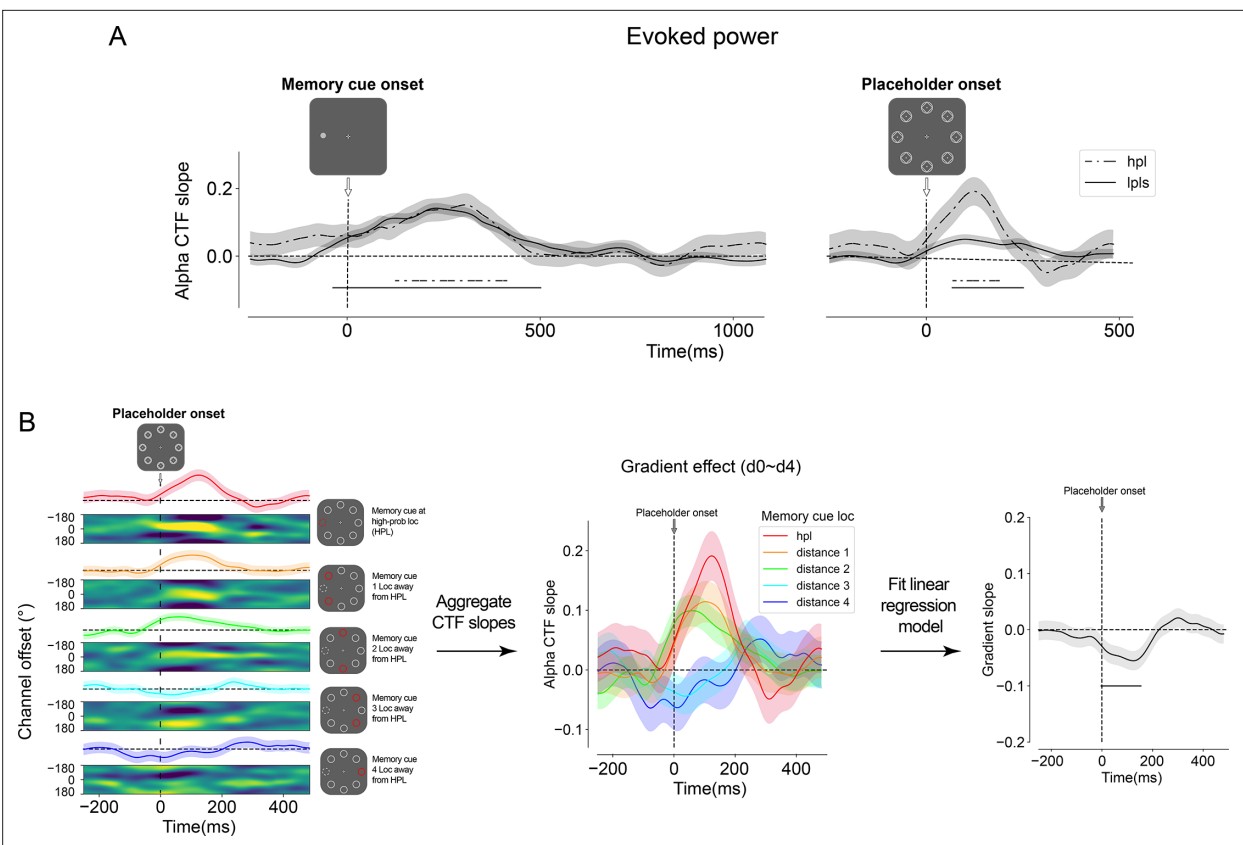

**Figure 5.** The gradient pattern of alpha channel tuning functions (CTFs) following the onset of the placeholder display. (**A**) Numerically larger CTFs were observed post-placeholder onset when the spatial memory cue matched the high-probability location compared to the low-probability locations. (**B**) The location-specific CTFs were categorized based on the relative position distance between the spatial memory cue and the high-probability location. To statistically evaluate the observed gradient pattern, a linear regression model was applied to the data points, followed by a permutation test on the slope of the regression model. Shaded areas depict bootstrapped SEM (N = 24), and time points with significant differences (cluster-based permutation test, p<0.05) in regression model slopes are indicated with horizontal black insets.

instead returned to baseline prior to placeholder onset (*Figure 4A*). These result patterns are in line with *van Moorselaar et al., 2018*, who also showed an attention shift and reallocation to the gradually relevant secondary search task. Critically, this information was again revived by the placeholder, an effect that was selective to evoked power (*Figure 4B*). Together, these findings are consistent with a model wherein the prospect of a secondary task during the memory maintenance interval shifted the memory from an activity-driven into an activity-silent state. Upon placeholder onset, the instantiated neutral input probed the system's underlying activity-silent state, revealing its spatially tuned priority weights (analogous to the pinging procedure). Despite a numerical trend, this pattern did not manifest in the total alpha power CTFs (see *Figure 4B*), indicating that the revived spatial selectivity may be attributed to the phase-locked information carried by the placeholder display.

The revival of the CTF by the placeholder display indicates that the hidden attentional priority map, which tracks the location of the memory cue, can still be reactivated despite the introduction of a secondary search task. To further investigate how this revival of the memory representation was modulated by distractor regularity, we analyzed the spatial selectivity indexed by the reconstructed CTF profiles as a function of the memory cue location (HPL-matched, LPLs-matched). As shown in *Figure 5A*, two key observations emerge. First, during the memory encoding period immediately following memory cue onset and the maintenance period prior to placeholder onset, both the HPL-matched and LPL-matched CTFs are robust (p<0.05) and highly overlap, indicating that, at this stage, the memory representation is unaffected by the spatial regularity of the distractor location. Second, upon placeholder onset, reliable CTF reconstruction is observed in both conditions (p<0.05), with the HPL-matched CTFs becoming numerically more pronounced than the LPL-matched CTFs. Although

this conditional difference was not statistically significant, the observed pattern suggests that memory cue locations overlapping with high-probability distractor locations were more strongly selected compared to those overlapping with LPL. This is inconsistent with the notion that distractor learning leads to proactive suppression of high-probability distractor locations, which would be expected to attenuate memory-specific revival at those locations.

To further explore this finding, we leveraged the observation that, as was also the case here, suppression often exhibits a spatial gradient. Specifically, we organized the evoked CTFs based on the distance from the memory location to the high-probability distractor location. As shown in *Figure 5B*, in line with a spatial gradient, the resulting CTF profiles varied depending on their distances to the high-probability distractor location. To evaluate the reliability of this observed gradient pattern in spatial selectivity, we employed a linear regression model to fit the tuning profiles. A cluster-based permutation test conducted on the slopes of the regression model identified a reliable gradient (p<0.05) centered around the HPL following placeholder onset. Critically, this gradient reflected an increase of spatial selectivity centered at the high-probability distractor location rather than an attenuation, suggesting that immediately following placeholder onset the to-be-suppressed location was at least transiently attended before being suppressed during visual search in the subsequent display.

## Discussion

The aim of the current study was to examine whether learned distractor suppression operates proactively (before the first shift of attention) or whether it is initiated reactively, following attentional selection (i.e. reactive suppression). To this end, we introduced a variant of the additional singleton task with a high-probability distractor location in the maintenance period of a spatial working memory task. At the behavioral level, responses were faster when the distractor occurred at the HPL, but slower when the target occurred at this location, indicative of generic spatial suppression, an effect that was characterized by a discernible spatial gradient centered at the high-probability distractor location. Yet, critically, at the neural level, there was no evidence that this spatial suppression attenuated processing at the high-probability distractor location in anticipation of a new search episode. The neutral placeholder display, serving as a ping, presented prior to search display onset revived the reconstruction of the CTF profile of the current memory representation. Instead of a model that assumes that the HPL was proactively suppressed, the revival of the CTF profile within the alpha-band showed the largest tuning at the learned suppressed location. Moreover, the modulation of the memory-tuned CTF profile was characterized by a spatial gradient that was centered over the HPL. These findings suggest that suppression is engaged following the initial selection of a location, indicating a tendency toward a reactive suppression mechanism within the specific context of our study. This implies that suppression may occur after attentional enhancement rather than being proactively implemented.

Within the realm of learned distractor suppression, an ongoing debate centers around the question of whether, and precisely when, visual distractors can be proactively suppressed. Such debates pertain to the multifaceted nature of distractor suppression acquired through learning, which can occur at the level of feature-based suppression (as proposed by the signal suppression hypothesis; *Gaspelin et al., 2015*; *Gaspelin and Luck, 2018a*; *Gaspelin and Luck, 2018b*; *Stilwell et al., 2019*) or spatial-based suppression (e.g. *Wang and Theeuwes, 2018b*; *Wang and Theeuwes, 2018a*). Specifically, proactive evidence for learned *spatial* distractor suppression is primarily based on findings showing that the behavioral benefit observed when distractors appear with a higher probability at a given location is accompanied by a probe detection cost (measured via dot offset detection) at the high-probability distractor location (*Huang et al., 2022*; *Huang et al., 2023*; *Huang et al., 2021b*). As also pointed out by *Chang et al., 2023*, however, this assumption is largely grounded in findings that rely on behavioral measures, leaving open the possibility that the suppression of a distractor location only occurs after it was initially attended. To bypass the ambiguity in behavioral measures, other studies have turned to measures of the brain in the window prior to search display onset. Studies investigating the preparatory bias in response to distractor regularities largely examined topographical modulations of alpha power, given its functional link to reduced cortical excitability. These studies, however, failed to find evidence in support of active preparatory inhibition as indexed via increased alpha power contralateral to the high-probability distractor location (*van Moorselaar et al., 2021a*; *van Moorselaar et al., 2020a*; *van Moorselaar and Slagter, 2019*; but see *Wang et al., 2019*). This absence of anticipatory tuning toward the to-be-suppressed location has given rise to the idea that distractor

learning operates by changing synaptic efficiency within these regions (*van Moorselaar and Slagter, 2020b*). Consistent with this, a recent study using rapid invisible frequency tagging demonstrated reduced neural excitability at the high-probability distractor location in the absence of any alpha-band modulations (*Ferrante et al., 2023*). However, the correlational approach in that study prevented a time-resolved analysis, leaving uncertainties about whether suppression was genuinely proactive or triggered by placeholder onset. Thus, while it is generally assumed that learned suppression is proactive in nature, the evidence in support of that notion remains equivocal.

The observed modulation of the revival of the CTF profile provides further insight into the neural mechanisms underlying more efficient distractor processing following statistical learning. Interestingly, rather than being suppressed, the revived CTFs were characterized by a spatial gradient centered over the HPL, with tuning being most pronounced at the to-be-suppressed location. This contrasts with the training phase, where tuning was homogeneous across all eight search locations. Together, this suggests that the putative priority map, initially tuned by maintaining a spatial location in WM, was reconfigured by statistical regularities across search displays to align with the imminent search. The dominant perspective in the field posits the existence of a spatial priority map that integrates various control signals, including sensory input, current goal states, and previous selection episodes, to encode the priority of individual locations (*Awh et al., 2012*; *Luck et al., 2021*; *Theeuwes et al., 2022*). If one assumes that the evoked response elicited by placeholder displays ping the landscape of this priority map (*Duncan et al., 2023*), the observed gradient in our study clearly argues against proactive suppression. Although this result is based on an exploratory analysis and should therefore be interpreted with caution, it is noteworthy that a recent behavioral study observed a similar pattern of results (*Chang et al., 2023*). In a modified version of the capture-probe paradigm, where the probe display onset was not preceded by a placeholder display, participants discriminated the orientation of a tilted bar presented at one of the search locations, revealing initial enhancement at the high-probability distractor location before suppression. However, two alternative explanations warrant consideration. First, one could argue that observed modulations in the revived CTFs do not provide insight into the mechanisms underlying distractor suppression but instead reflect changes in the memory representation itself, potentially triggered by the anticipation of the HPL in the search task. According to this view, the changes in the revived CTFs would be unrelated to how search performance (in particular, distractor suppression) was achieved. While this is theoretically possible, we believe it to be unlikely. Memory performance (recall) did not vary as a function of the cue's distance from the HPL (see *Figure 2—figure supplement 1*), whereas the revived CTFs did, indicating that these changes likely reflect contributions from both tasks. Additionally, distractor learning typically occurs without conscious awareness (*Gao and Theeuwes, 2022*; *Wang and Theeuwes, 2018b*). It is difficult to conceive how such unconscious processes could produce *anticipatory* effects in the memory task and selectively modulate the representation of the consciously remembered memory cue. Second, the apparent lack of suppression and the presence of a pronounced tuning at the high-probability distractor location could actually reflect a proactive mechanism that manifests in a way that seems reactive due to the dual-task nature of our experiment. Expectations about upcoming distractor information may also result in anticipatory distractor tuning such that the actual presentation of a distractor elicits a weaker response (i.e. prediction error), thereby reducing distractor interference (*van Moorselaar and Slagter, 2020b*). In this interpretation, expectations that actually serve to silence information processing will result in enhanced tuning toward the HPL when combined with a spatial memory at that location. Irrespective however, the current results clearly argue against a model wherein statistical learning operates by downregulating weights at the to-be-suppressed location.

While many studies, including the current one, have attempted to dissociate between proactive and reactive suppression mechanisms, it is plausible that these processes are not mutually exclusive and may often contribute concurrently to observed attentional effects. For example, eye tracking studies with a spatial distractor imbalance have revealed a more nuanced picture of attentional suppression. A study by *Wang et al., 2019*, showed that fewer saccades landed at the suppressed location than any other location, consistent with the notion that there was less proactive oculomotor activity at the suppressed location resulting in fewer saccades to that location. Critically however, there was also evidence for faster disengagement from the high-probability distractor location than the low-probability distractor location, suggesting the operation of a reactive mechanism where suppression is followed by rapid oculomotor disengagement (*Sauter et al., 2021*).

A novel aspect of the current study is that visual pings, here in the shape of a neutral placeholder display, can effectively unveil spatial memories hidden from the ongoing EEG signal. Recently, we demonstrated that the otherwise, invisible attentional priority map induced by a spatial imbalance of target probability across trials could also be revealed by the 'pinging' technique in conjunction with multivariate pattern analysis and EEG (*Duncan et al., 2023*). However, while it is well established that early ERP components are enhanced in response to visual stimulation at memorized locations (*Awh et al., 2000*), to date the pinging technique has exclusively been used to reveal feature-specific information within working memory (*Wolff et al., 2015*; *Wolff et al., 2017*). Despite initial enthusiasm within this field, there is an ongoing debate surrounding the precise mechanisms underlying the ping effect, particularly whether it reactivates latent networks or merely amplifies existing, yet below-threshold representations within ongoing neural activity (*Barbosa et al., 2021*). Additionally, skepticism arises from demonstrations that measurable neural activity often underlies working memory maintenance (*Schneegans and Bays, 2017*). These two, not necessarily exclusive scenarios, were also evident in the present study. During the training phase, where the memory task was the only task at hand, CTFs continuously tracked the position of the spatial memory cue. However, the same reconstruction returned to baseline when the model was applied to the test session that incorporated a search task during the maintenance interval. This dissociation aligns intriguingly with a study by *van Moorselaar et al., 2018*, where continuous reconstruction of spatial memory was disrupted during a secondary task introduced in the maintenance interval. This suggests the possibility that under dual-task conditions memories might be strategically offloaded to a mechanism relying on weak neural activity or even one not dependent on sustained neural firing at all, such as an activity-silent representation facilitated by synaptic plasticity or long-term memory. Regardless of the precise underlying mechanism, our results demonstrate in a compelling way that the content of a hidden spatial representation can be revived by flooding the visual system with sensory input. This demonstrates that the pinging technique can not only be used to investigate feature-based working memory but also the dynamics of spatial memories.

In summary, the present study is the first to show that a spatial memory representation can be reconstructed based on a ping, a neutral placeholder display, and used to infer how distractor suppression affects the priority map prior to search. Prior to search and in response to the ping, the tuning profile of the memorized location was most pronounced at the high-probability distractor location and exhibited a spatial gradient centered over that location. Although the exploratory nature of our findings should again be stressed, they may call for a reinterpretation of how learned suppression might take place. Instead of proactively suppressing specific locations, individuals may first direct attention to the location that they have implicitly learned to expect a distractor, enabling suppression of that location following rapid attentional disengagement.

## Methods
### Participants

In line with previous EEG experiments (*Duncan et al., 2023*; *Noonan et al., 2016*; *van Moorselaar et al., 2021a*; *van Moorselaar et al., 2023*; *van Moorselaar et al., 2020a*), we used a predetermined sample size of 24 participants. Participants ($N$=24, 22 females, 2 males, $M_{age}$ = 20.8, $SD_{age}$ = 2.9) were recruited from the research pool of the Vrije Universiteit Amsterdam. Participants received either course credits or a monetary reward (€35) for 3.5 hr of participation. To ensure data quality and maintain the predefined sample size, three participants were replaced. One participant was substituted due to a high number of trial removals during preprocessing (>30%). Another participant, whose accuracy fell below 2.5 standard deviations from the overall mean for the search task, was also replaced. The third participant was substituted because of poor memory recall, deviating more than 2.5 standard deviations from the overall mean. Written informed consent was obtained from all participants before the experiment. The study was approved by the Ethical Review Committee of the Faculty of Behavioral and Movement Sciences of Vrije Universiteit Amsterdam (protocol number: VCWE-2021-173) and was conducted following the guidelines of the Helsinki Declaration.

## Apparatus, task, and stimuli

The experiment was created on a Dell Precision 3640 Windows 10 computer equipped with an NVIDIA Quadro P620 graphics card in OpenSesame (*Mathôt et al., 2012*) using PsychoPy functionality (*Peirce, 2007*). Stimuli were presented on a 23.8-in ASUS XG248Q monitor with a 240 Hz refresh rate. Participants were seated in a dimly lit room, positioned at a distance of 70 cm from the monitor, with the aid of a chin rest to ensure stable head position. Throughout the experiment, participants' right eye movements were tracked by Eyelink 1000 (SR Research) eye tracker at a sample rate of 1000 Hz. Participants were given specific instructions to maintain their fixation on the central point during the experiment. Auditory feedback was provided whenever the gaze position deviated by more than 1.5° from the central point. All visual stimuli were presented against a light gray background with RGB values of 224/224/224. Henceforth, the colors of the stimuli were specified as RGB values.

The spatial working memory task was adapted from *Foster et al., 2016*, and required participants to memorize the angular location of a circle stimulus (0.9° in radius, 196/196/196) positioned 4.2° away from the central fixation marker (0.35° in radius). This fixation marker was designed as a combination of a bull's eye and crosshair, a feature known to improve stable fixation (*Thaler et al., 2013*). The angular location of the memory cue was sampled with equal probability from one of eight location bins spanning 0°–315°, with jitter added (–22.5° –22.5°) to cover all possible locations to avoid categorical coding. At test, participants were required to report the memorized location via a mouse by clicking on a ring (4.2° in radius).

The intermediate visual search task was modeled after the additional singleton paradigm (*Theeuwes, 1992*). Participants were instructed to search for a unique shape (i.e. target) within a further homogeneous display and report the orientation of the line within the target shape. The search array consisted of eight evenly spaced items in a circular configuration (4.2° in radius) around central fixation. Each item within the search array contained either a vertical or horizontal white bar (0.1° × 1°, 255/255/255). The target could either be a diamond among seven circles or vice versa. Due to the differences in geometric shape, we chose to make the diamond slightly larger (2.3° × 2.3°) than the circle (2° in diameter) to ensure a better visual match in overall size appearance (*Duncan et al., 2023*; *Huang et al., 2023*). In 74% of the trials, a distractor was present: one of the non-target items was a color singleton (either red (255/0/0) or green (0/131/0)). In these trials, the distractor was more likely (65%) to be placed at one location, called the HPL, than at the other locations (5% at each of these seven locations), to induce statistically learned suppression of the HPL. The HPL remained constant throughout the experiment and was counterbalanced across participants. The target color, shape, and line orientation within the target were randomized on each trial, and the target location was equally likely across all locations. Please note that although the colors were not equiluminant, the target and distractor colors were randomized across trials such that roughly half the trials had a red distractor, and half had a green distractor. This randomization process should help mitigate any systematic biases this may cause.

The placeholder display, which served as a visual ping, consisted of eight gray shapes (128/128/128), each created by superimposing a diamond shape onto a circle shape. Importantly, the spatial locations that were occupied by the placeholder and the search array were spatially matched with the eight position bins in the spatial working memory task.

## Design and procedure

The current study followed a structured protocol comprising two phases, a training phase, which served as an independent localizer to train the encoding model (details below), and a subsequent test phase combining the spatial working memory task with the visual search task. During the training phase, each trial began with a blank display for a randomly jittered duration of 200–400 ms, followed by a fixation dot for 250 ms. Participants were explicitly instructed to maintain their gaze on the fixation dot as long as it was visible. Subsequently, a memory cue was presented on the screen, and participants were instructed to remember its location until the end of the trial. The memory cue disappeared after 200 ms, leaving only the fixation dot visible for a duration that ranged randomly between 900 and 1100 ms. Next, a placeholder display appeared for a randomly varying duration between 500 and 750 ms. Finally, a memory test display was presented until a response was made by participants.

In the test phase, the trial structure closely resembled that of the training phase, with two important distinctions: First, following the placeholder display, a search display was presented for a maximum

duration of 2000 ms or until a response was made. Second, on a small subset of trials (13%), the spatial memory cue display was replaced by a display containing a yellow fixation, signaling participants that they only needed to perform the visual search task. [The goal of these no-memory trials was to track learned attentional priority in the absence of a top-down attentional set. The resulting model, however, failed to capture any reliable spatial tuning relative to the HPL, leading us to omit reporting this particular outcome].

The training phase consisted of 10 blocks, each containing 80 trials, while the test phase comprised of 10 blocks of 92 trials. Between blocks, participants were given the opportunity of a short break, during which feedback on the mean memory recall deviation and search performance (in the test phase) for both the latest block and all finished blocks was provided. After the completion of the training phase, participants were given a 5 min break before proceeding to the test phase. Before entering the training phase, participants performed 16 practice trials randomly drawn from a training block. To proceed, they had to meet a criterion of an average memory recall deviation of less than 13°. Likewise, preceding the test phase, participants underwent 48 practice trials randomly drawn from a testing block. To qualify for the test phase, they needed to achieve at least 65% accuracy in the search task and verbally confirm their understanding of the task goals with the experimenter. It is important to emphasize that during the test phase, the location of the memory cue was not contingent on the location of the singleton in the search task (i.e. these two tasks were independent of each other).

## Behavioral analysis

All data were preprocessed using Python, and statistical analysis was done using R (*R Development Core Team, 2020*). Conditional mean RTs and accuracy of the visual search task were analyzed with repeated measures ANOVA, followed by planned comparisons with paired t-tests. For RT analyses, we excluded incorrect responses and RTs <200 ms. For each participant, we also excluded RTs that exceeded the ±2.5 standard deviation from the overall mean RT (collapsed across conditions). The exclusion of incorrect responses and data trimming resulted in an average loss of 10.5% of trials. Participants whose accuracy or memory deviation fell above or below 2.5 SD from the overall mean were replaced during the data collection phase (see *Participants*). p-Values were Greenhouse-Geisser corrected in cases where the assumption of sphericity was violated and were corrected with the Holm-Bonferroni method for multiple-level comparisons. In cases of nonsignificant findings, we also provided the Bayes factor ($BF_{01}$) to support the null model.

## EEG recording and preprocessing

EEG data were collected using a 64-electrode cap with electrodes placed according to the 10-10 system (Biosemi ActiveTwo system; https://www.biosemi.com/). To monitor eye movements, in case of missing eye tracker data, vertical and horizontal electrooculogram (VEOG/HEOG) signals were recorded via external electrodes placed ~2 cm above and below the right eye, and ~1 cm lateral to the external canthi. Two additional electrodes were placed on the left and right earlobe for offline reference. Electrode impedances were kept below 20 kΩ. Signals were amplified (100 Hz low-pass filter, 0.16 Hz high-pass filter; ActiveTwo AD-box, ActiveView) and sampled at 512 Hz.

EEG data were preprocessed using a customized Python script and the MNE package (*Gramfort et al., 2013*). During preprocessing, the data were re-referenced to the average of the left and the right earlobe and high-pass filtered using a zero-phase 'firwin' filter at 0.01 Hz to remove slow drifts. Malfunctioning electrodes detected during recording were temporarily removed in offline analysis such that subsequent preprocessing steps were not influenced by these electrodes. The continuous data were then epoched from 750 to 1600 ms relative to the memory cue onset and from 750 to 1000 ms relative to the placeholder display onset, with the windows of interest being –250 to 1100 ms and –250 to 500 ms, respectively (centered around memory cue and placeholder display onset). Eyeblink components were removed after performing an independent components analysis as implemented in MNE (method = 'picard') on 1 Hz filtered epochs. We used an automatic trial-rejection procedure on the EEG signal to remove noise-contaminated epochs. Specifically, the EEG signal was further processed by applying a 110–140 Hz band-pass filter to capture muscle activity and transform it into z-scores. A subject-specific z-score threshold was determined based on the within-subject variance of z-scores within the windows of interest (*de Vries et al., 2017*; *Duncan et al., 2023*). To minimize false alarms, instead of immediately removing epochs that exceeded the z-score threshold, an

iterative procedure was employed. For each marked epoch, the five electrodes contributing most to the accumulated z-score within the time period containing the marked artifact were identified. These electrodes were then interpolated one by one using spherical splines (*Perrin et al., 1989*). After each interpolation, the epoch was checked to see if it still exceeded the z-score threshold. Epochs that still exceeded the threshold after the iterative interpolation were dropped which led to an average loss of 9.1% of all trials (range 0.1–15.3%) for epochs time-locked to memory cue onset and an average loss of 6.1% of all trials (range 0–12.3%) for epochs time-locked to placeholder onset. Lastly, malfunctioning electrodes were interpolated using spherical splines (*Perrin et al., 1989*).

Samples of the eye positions were aligned with the EEG data during offline analysis and converted to visual degrees that deviated from the central fixation point. To prevent potential confounds in interpreting the results, epochs were excluded if the gaze position exceeded 1.2° of the central fixation point anytime during the time range of –100 to 500 ms relative to the memory cue onset or during the time range of –200 to 300 ms relative to the placeholder onset. In case of missing Eyelink data, epochs were removed when detecting a sudden increase in HEOG amplitude via an algorithm with a window size of 200 ms, a step of 10 ms, and a threshold of 15 µV. In total, 6.8% of the cleaned data for memory epochs (range 0.4–18.7%) and 3.2% of the cleaned data for placeholder epochs (range 0–8.2%) were excluded due to the detection of eye movements.

## Time-frequency analysis

We performed time-frequency analysis to measure total and evoked alpha power. To isolate power in the alpha frequency band, we filtered the artifact-free EEG signals with a fifth-order Butterworth band-pass filter (8–13 Hz) implemented within MNE. We then applied a Hilbert transform to the filtered data to obtain the complex analytic signal. Evoked power was calculated by averaging the complex analytic signals obtained from the Hilbert transform across trials, and then squaring the averaged complex magnitude to obtain power. Thus, evoked power captures activity that is phase-locked to stimulus onset because only activity with consistent phase across trials remains when the complex analytic signal is averaged. In contrast, total power was calculated by first squaring the complex magnitude to obtain power for each trial, and then averaging power across trials. Thus, total power captures all alpha activity, regardless of its phase relationship to stimulus onset. Because distinguishing between evoked and total power requires averaging across trials, we partitioned our data into sets (see *Training and test data*), and we calculated evoked and total power for each memory cue location by averaging (as described above) across trials for each memory cue location. All power time series were then smoothed with a sliding window approach (window size = 8 samples [15.62 ms], step = 1). Subsequently, a principal component analysis (n_components = 16) was performed on the smoothed data of all electrodes to enhance the signal-to-noise ratio.

## Inverted encoding model

We used an IEM to reconstruct location-selective CTFs from the pattern of alpha-band power across the scalp (*Foster et al., 2016*). This analysis assumes that power (evoked or total, see *Time-frequency analysis*) measured at each electrode reflects the weighted sum of eight spatial channels (i.e. neuronal populations), each tuned to a different angular location (*Foster et al., 2016*). We modeled the response profile of each spatial channel across angular locations using a half-sinusoid:

$$R = sin\left(0.5\theta\right)^{7}$$

where $\theta$ represents angular locations ranging from 0° to 359° and $R$ is the response of the spatial channel. This response profile was shifted circularly for each channel such that the peak response of each spatial channel was centered at eight equally spaced polar angles (0°, 45°, 90°, etc.). These basis functions specified the predicted responses of each of the channels for each angular location.

We applied an IEM procedure to power (evoked and total) at each time point. Having partitioned our data into independent training and test sets (see *Training and test data*), the analysis proceeded in in two stages: training and test. In the training stage, training data ($B_1$) were used to estimate a weight matrix that reflects the relatively contribution of each spatial channel to power measured at each electrode. Let $B_1$ ($m$ electrodes×$n_1$ observations) be the power at each electrode for each observation in the training set, $C_1$ ($k$ channels×$n_1$ observations) be the predicted response of each spatial channel (determined by the basis functions) for each measurement, and $W$ ($m$ electrodes×$k$ channels)

be a weight matrix that characterizes a linear mapping from 'channel space' to 'electrode space'. The relationship between $B_1$, $C_1$, and $W$ can be described by a GLM of the form:

$$B_1 = WC_1$$

The weight matrix ($W$) was estimated via the least squares solution using the Python function np.linalg.lstsq($C_1$, $B_1$).

In the test stage, we inverted the model to transform the observed test data $B_2$ ($m$ electrodes×$n_2$ observations) into estimated channel responses, $C_2$ ($k$ channels×$n_2$ observations), using the estimated weight matrix ($\hat{W}$) with the Python function, np.linalg.lstsq($W.T$, $B_2.T$). We circularly shifted the resulting channel responses ($C_2$) so that they were centered on the common center (0°), corresponding to the location of the memory cue. The derived channel responses (8 channels×8 location bins) were then used for the following analyses: (a) calculating individual CTFs based on each of the eight physical location bins (e.g. *Figure 3C and D*); (b) grouping responses according to the distance between each physical location and the high-probability distractor location to calculate distance CTFs (e.g. *Figure 5*); and (c) averaging across location bins to represent the general strength of spatial selectivity in tracking the memory cue, irrespective of its specific location (e.g. *Figure 3A and B*).

## Training and test data

For all our IEM analyses, we randomly partitioned our data into three independent sets, with two serving as the training data ($B_1$), and the third serving as the test data ($B_2$). We discarded some trials to equate the number of trials for each memory cue location. For each set, we averaged across the trials for each memory cue location to calculate power (both evoked and total, see Time-frequency analysis). In our localizer analysis (*Figure 3*), we reconstructed CTFs during the localizer task only. Thus, data from the localizer task was randomly partitioned into three sets, with two of these sets serving as the training data ($B_1$), while the remaining set served as the test data ($B_2$). In our main analyses (i.e. cross-session analysis in *Figures 4 and 5*), we used data from the training phase/localizer task to fit the IEM, and then reconstructed CTFs during the test phase task. To this end, we randomly partitioned data from our localizer task into two sets, which served as training data ($B_1$), and the data from the test phase task served as the test data ($B_2$). In both analyses, we used an iterative approach to reduce noise in the estimated CTFs. For each of 100 iterations, we randomly partitioned the data into training and test (as described above), and we averaged the resulting CTFs across iterations.

## Statistical analysis

To quantify the spatial selectivity of the CTFs, we employed a linear polynomial fitting approach to estimate the slopes of the CTFs, where larger slopes indicate greater tuning for the location of the memory cue would signify spatial selectivity, while negative slopes would indicate deviation from spatial selection (i.e. suppression). These slope values were then subjected to statistical analysis to assess the validity of the reconstructed CTFs and conditional differences, using a cluster-based one-sample paired t-test combined with Monte Carlo randomization (the *permutation_cluster_1samp_test* function within the MNE package). This nonparametric statistical testing considers the data's temporal correlation structure and controls for multiple comparison problems (*Maris and Oostenveld, 2007*). A new dataset was randomly resampled from the observed data in each iteration. The sign of the resample data was randomly flipped, and clusters were identified if the t-values of the adjacent data points exceeded the threshold. The cluster with the largest sum of t-values was retained. This process was repeated 1024 times, yielding a permutation distribution from the retained clusters. The clusters obtained from the veridical data were then compared with the form distribution. The slope or the conditional difference was deemed statistically reliable if the cluster exceeded the 95th percentile of the distribution.

## Acknowledgements

We would like to thank Zhenzhen Xu and Ya Gao for their invaluable assistance in data collection. We also appreciate Johannes Fahrenfort for his insightful discussion on data analysis.

## Additional information

### Funding

| Funder | Grant reference number | Author |
| --- | --- | --- |
| European Research Council | 833029 | Jan Theeuwes |
| China Scholarship Council | 201908440284 | Changrun Huang |

The funders had no role in study design, data collection and interpretation, or the decision to submit the work for publication.

### Author contributions

Changrun Huang, Conceptualization, Data curation, Formal analysis, Writing – original draft, Writing – review and editing; Dirk van Moorselaar, Conceptualization, Formal analysis, Supervision, Writing – original draft, Writing – review and editing; Joshua Foster, Conceptualization, Formal analysis, Writing – review and editing; Mieke Donk, Conceptualization, Supervision, Writing – original draft, Writing – review and editing; Jan Theeuwes, Conceptualization, Supervision, Funding acquisition, Writing – original draft, Writing – review and editing

### Author ORCIDs

Changrun Huang (iD) https://orcid.org/0000-0002-1627-0887

### Ethics

Written informed consent was obtained from all participants before the experiment. The study was approved by the Ethical Review Committee of the Faculty of Behavioral and Movement Sciences of Vrije Universiteit Amsterdam (protocol number: VCWE-2021-173) and was conducted following the guidelines of the Helsinki Declaration.

Reviewer #1 (Public review): https://doi.org/10.7554/eLife.98304.3.sa1
Reviewer #2 (Public review): https://doi.org/10.7554/eLife.98304.3.sa2
Reviewer #3 (Public review): https://doi.org/10.7554/eLife.98304.3.sa3
Author response https://doi.org/10.7554/eLife.98304.3.sa4

## Additional files

### Supplementary files

MDAR checklist

### Data availability

All data generated or analyzed during this study have been deposited in the Open Science Framework (OSF) and are publicly available at https://osf.io/ub5gq/.

The following dataset was generated:

| Author(s) | Year | Dataset title | Dataset URL | Database and Identifier |
| --- | --- | --- | --- | --- |
| Huang C, van Moorselaar D, Foster JJ, Donk M, Theeuwes J | 2025 | Pinging the Hidden Attentional Priority Map: Suppression Needs Attention | https://osf.io/ub5gq/ | Open Science Framework, ub5gq |

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
