## [Editor Report · eLife Assessment]

This **important** study uses recently developed EEG analysis methods to investigate spatial distractor suppression in a combined visual search/working memory task. The reported results are **compelling**, although they are open to multiple interpretations. The study will be of interest to cognitive neuroscientists and psychologists working on visual attention and memory.

---

## [Referee Report · Reviewer #1 (Public review)]

Summary:

The authors tested whether learning to suppress (ignore) salient distractors (e.g., a lone colored nontarget item) via statistical regularities (e.g., the distractor is more likely to appear in one location than any other) was proactive (prior to paying attention to the distractor) or reactive (only after first attending the distractor) in nature. To test between proactive and reactive suppression the authors relied on a recently developed and novel technique designed to "ping" the brain's hidden priority map using EEG inverted encoding models. Essentially, a neutral stimulus is presented to stimulate the brain, resulting in activity on a priority map which can be decoded and used to argue when this stimulation occurred (prior to or after attending a distracting item). The authors found evidence that despite learning to suppress the high probability distractor location, the suppression was reactive, not proactive in nature.

Overall, the manuscript was well-written, tests a timely question, and provides novel insight into a long-standing debate concerning distractor suppression.

The authors provided a thorough rebuttal and addressed the previous critiques and concerns.

Strengths (in no particular order):

(1) The manuscript is well-written, clear, and concise (especially given the complexities of the method and analyses).

(2) The presentation of the logic and results is clear and relatively easy to digest.

(3) This question concerning whether location-based distractor suppression is proactive or reactive in nature is a timely question.

(4) The use of the novel "pinging" technique is interesting and provides new insight into this particularly thorny debate over the mechanisms of distractor suppression.

Weaknesses (in no particular order):

After revision, the prior weaknesses have been largely addressed.

---

## [Referee Report · Reviewer #2 (Public review)]

Summary:

The authors investigate the mechanisms supporting learning to suppress distractors at predictable locations, focusing on proactive suppression mechanisms manifesting before the onset of a distractor. They used EEG and inverted encoding models (IEM). The experimental paradigm alternates between a visual search task and a spatial memory task, followed by a placeholder screen acting as a 'ping' stimulus -i.e., a stimulus to reveal how learned distractor suppression affects hidden priority maps. Behaviorally, their results align with the effects of statistical learning on distractor suppression. Contrary to the proactive suppression hypothesis, which predicts reduced memory-specific tuning of neural representations at the expected distractor location, their IEM results indicate increased tuning at the high-probability distractor location following the placeholder and prior to the onset of the search display.

Strengths:

Overall, the manuscript is well-written and clear, and the research question is relevant and timely, given the ongoing debate on the roles of proactive and reactive components in distractor processing. The use of a secondary task and EEG/IEM to provide a direct assessment of hidden priority maps in anticipation of a distractor is, in principle, a clever approach. The study also provides behavioral results supporting prior literature on distractor suppression at high-probability locations.

Weaknesses:

In response to my comments during the first review, the authors have clarified and further discussed several methodological aspects, limitations, and alternative interpretations, tempering some of their claims and, overall, improving the manuscript. These involved mostly broadening the introduction and discussion of the putative mechanisms in distractor suppression, evaluating alternative explanations due to the dual-task design, clarifying methodological details regarding the inverted encoding model, and discussing the possibility that proactive suppression might actually require enhanced tuning toward the expected feature. While, to some degree, the results may still remain open to alternative explanations, the study, in its current form, presents an interesting paradigm and promising findings that will undoubtedly be useful for future research. I therefore have no major remaining comments.

---

## [Referee Report · Reviewer #3 (Public review)]

Summary:

In this experiment, the authors use a probe method along with time-frequency analyses to ascertain the attentional priority map prior to a visual search display in which one location is more likely to contain a salient distractor. The main finding is that neural responses to the probe indicate that the high probability location is attended, rather than suppressed, prior to the search display onset. The authors conclude that suppression of distractors at high probability locations is a result of reactive, rather than proactive, suppression.

Strengths:

This was a creative approach to a difficult and important question about attention. The use of this "pinging" method to assess the attentional priority map has a lot of potential value for a number of questions related to attention and visual search. Here as well, the authors have used it to address a question about distractor suppression that has been the subject of competing theories for many years in the field. The authors have also conducted additional behavioral analyses to examine the relationship between memory and search. The paper is well-written, and the authors have done a good job placing their data in the larger context of recent findings in the field.

Weaknesses:

The authors addressed a number of weaknesses in a thorough revision during the review process. The present study raises important questions for future research - this is not a weakness, since one study cannot answer all questions, but points to the importance of the questions raised by this study and the value of additional future research in the area.

---

## [Author Response]

The following is the authors’ response to the original reviews.

**Public Reviews:**

**Reviewer #1 (Public Review):**
Summary:The authors tested whether learning to suppress (ignore) salient distractors (e.g., a lone colored nontarget item) via statistical regularities (e.g., the distractor is more likely to appear in one location than any other) was proactive (prior to paying attention to the distractor) or reactive (only after first attending the distractor) in nature. To test between proactive and reactive suppression the authors relied on a recently developed and novel technique designed to "ping" the brain's hidden priority map using EEG inverted encoding models. Essentially, a neutral stimulus is presented to stimulate the brain, resulting in activity on a priority map which can be decoded and used to argue when this stimulation occurred (prior to or after attending to a distracting item). The authors found evidence that despite learning to suppress the high probability distractor location, the suppression was reactive, not proactive in nature.Overall, the manuscript is well-written, tests a timely question, and provides novel insight into a long-standing debate concerning distractor suppression.Strengths (in no particular order):(1) The manuscript is well-written, clear, and concise (especially given the complexities of the method and analyses).(2) The presentation of the logic and results is mostly clear and relatively easy to digest.(3) This question concerning whether location-based distractor suppression is proactive or reactive in nature is a timely question.(4) The use of the novel "pinging" technique is interesting and provides new insight into this particularly thorny debate over the mechanisms of distractor suppression.Weaknesses (in no particular order):(1) The authors tend to make overly bold claims without either (A) mentioning the opposing claim(s) or (B) citing the opposing theoretical positions. Further, the authors have neglected relevant findings regarding this specific debate between proactive and reactive suppression.(2) The authors should be more careful in setting up the debate by clearly defining the terms, especially proactive and reactive suppression which have recently been defined and were more ambiguously defined here.(3) There were some methodological choices that should be further justified, such as the choice of stimuli (e.g., sizes, colors, etc.).(4) The figures are often difficult to process. For example, the time courses are so far zoomed out (i.e., 0, 500, 100 ms with no other tick marks) that it makes it difficult to assess the timing of many of the patterns of data. Also, there is a lot of baseline period noise which complicates the interpretations of the data of interest.(5) Sometimes the authors fail to connect to the extant literature (e.g., by connecting to the ERP components, such as the N2pc and PD components, used to argue for or against proactive suppression) or when they do, overreach with claims (e.g., arguing suppression is reactive or feature-blind more generally).

We thank the reviewer for their insightful feedback and have made several adjustments to address the concerns raised. To provide a balanced discussion, we tempered our claims about suppression mechanisms and incorporated additional references to opposing theoretical positions, including the signal suppression hypothesis, while clarifying the definitions of proactive and reactive suppression based on recent terminology (Liesefeld et al., 2024). We justified methodological choices, such as the slight size differences between stimuli to achieve perceptual equivalence and the randomization of target and distractor colors to mitigate potential luminance biases. We have revised our figure to enhance figure clarity. Lastly, while our counterbalanced design precluded reliable ERP assessments (e.g., N2pc, PD), we discussed their potential relevance for future research and ensured consistency with the broader literature on suppression mechanisms.

**Reviewer #2 (Public Review):**
Summary:The authors investigate the mechanisms supporting learning to suppress distractors at predictable locations, focusing on proactive suppression mechanisms manifesting before the onset of a distractor. They used EEG and inverted encoding models (IEM). The experimental paradigm alternates between a visual search task and a spatial memory task, followed by a placeholder screen acting as a 'ping' stimulus -i.e., a stimulus to reveal how learned distractor suppression affects hidden priority maps. Behaviorally, their results align with the effects of statistical learning on distractor suppression. Contrary to the proactive suppression hypothesis, which predicts reduced memory-specific tuning of neural representations at the expected distractor location, their IEM results indicate increased tuning at the high-probability distractor location following the placeholder and prior to the onset of the search display.Strengths:Overall, the manuscript is well-written and clear, and the research question is relevant and timely, given the ongoing debate on the roles of proactive and reactive components in distractor processing. The use of a secondary task and EEG/IEM to provide a direct assessment of hidden priority maps in anticipation of a distractor is, in principle, a clever approach. The study also provides behavioral results supporting prior literature on distractor suppression at high-probability locations.Weaknesses:(1) At a conceptual level, I understand the debate and opposing views, but I wonder whether it might be more comprehensive to present also the possibility that both proactive and reactive stages contribute to distractor suppression. For instance, anticipatory mechanisms (proactive) may involve expectations and signals that anticipate the expected distractor features, whereas reactive mechanisms contribute to the suppression and disengagement of attention.

This is an excellent point. Indeed, while many studies, including our own, have tried to dissociate between proactive and reactive mechanisms, as if it is one or the other, the overall picture is arguably more nuanced. We have added a paragraph to the discussion on page 19 to address this. At the same time, (for more details see our responses to your comments 3 and 5), we have added a paragraph where we provide an alternative explanation of the current data in the light of the dual-task nature of our experiment.

(2) The authors focus on hidden priority maps in pre-distractor time windows, arguing that the results challenge a simple proactive view of distractor suppression. However, they do not provide evidence that reactive mechanisms are at play or related to the pinging effects found in the present paradigm. Is there a relationship between the tuning strength of CTF at the high-probability distractor location and the actual ability to suppress the distractor (e.g., behavioral performance)? Is there a relationship between CTF tuning and post-distractor ERP measures of distractor processing? While these may not be the original research questions, they emerge naturally and I believe should be discussed or noted as limitations.

Thank you for raising these important points. While CTF slopes have been shown to provide spatially and temporally resolved tracking of covert spatial attention and memory representations at the group level, to the best of our knowledge, no study to date has found a reliable correlation between CTFs and behavior. Moreover, the predictive value of the learned suppression effect, while also highly reliable at the group level, has been proven to be limited when it comes to individual-level performance (Ivanov et al. 2024; Hedge et al., 2018). Nevertheless, based on your suggestion, we explored whether there was a correlation between the averaged gradient slope within the time window where the placeholder revived the memory representation and the average distance slope in reaction times for the learned suppression effect. This correlation was not significant (*r* = .236, *p* = 0.267), which, considering our sample size and the reasons mentioned earlier, is not particularly surprising. Given that our sample size was chosen to measure group level effects, we decided not to include individual differences analysis it in the manuscript.

Regarding the potential link between the CTF tuning profile and post-distractor ERP measures like N2pc and Pd, our experimental design presented a specific challenge. To reliably assess lateralized ERP components like N2pc or Pd the high probability location must be restricted to static lateralized positions (e.g., on the horizontal midline). Our counterbalanced design (see also our response to comment 9 by reviewer 1), which was crucial to avoid bias in spatial encoding models, precluded such a targeted ERP analysis.

(3) How do the authors ensure that the increased tuning (which appears more as a half-split or hemifield effect rather than gradual fine-grained tuning, as shown in Figure 5) is not a byproduct of the dual-task paradigm used, rather than a general characteristic of learned attentional suppression? For example, the additional memory task and the repeated experience with the high-probability distractor at the specific location might have led to longer-lasting and more finely-tuned traces for memory items at that location compared to others.

Thank you for raising these important points. Indeed, a unique aspect of our study that sets it apart from other studies, is that the effects of learned suppression were not measured directly via an index of distractor processing, but rather inferred indirectly via tuning towards a location in memory. The critical assumption here, that we now make explicit on page 18, is that various sources of attentional control jointly determine the priority landscape, and this priority landscape can be read out by neutral ping displays. An alternative however, as suggested by the reviewer, is that memory representations may have been sharper when they remembered location was at the high probability distractor location. We believe this is unlikely for various reasons. First, at the behavioral level there was no evidence that memory performance differed for positions overlapping high and low probability distractor locations (also see our response to reviewer 3 minor comment 4). Second, there was no hint whatsoever that the memory representation already differed during encoding or maintenance (This is now explicitly indicated in the revised manuscript on page 14), which would have been expected if the spatial distractor imbalance modulated the spatial memory representations.

Nevertheless, as discussed in more detail in response to comment 5, there is an alternative explanation for the observed gradient modulation that may be specific to the dual nature of our experiment.

(4) It is unclear how IEM was performed on total vs. evoked power, compared to typical approaches of running it on single trials or pseudo-trials.

Thank you for pointing out that our methods were not clear. We did not run our analysis on single trials because we were interested in separately examining the spatial selectivity of both evoked alpha power (phase locked activity aligned with stimulus onset) and total alpha power (all activity regardless of signal phase). It is only possible to calculate evoked and total power when averaging across trials. Thus, when we partitioned the data into sets for the IEM analysis, we averaged trials for each condition/stimulus location to obtain a measurement of evoked and total power each condition for each set. This is the same approach used in previous work (e.g. Foster et al., 2016; van Moorselaar et al., 2018).

We reviewed our method section and can see why this was unclear. In places, we had incorrectly described the dimensions of training and test data as electrodes x trials. To address this, we’ve rewritten the “Time frequency analysis”, “Inverted encoding model” sections, and added a new “Training and test data” section. We hope that these sections are easier to follow.

(5) Following on point 1. What is the rationale for relating decreased (but not increased) tuning of CTF to proactive suppression? Could it be that proactive suppression requires anticipatory tuning towards the expected feature to implement suppression? In other terms, better 'tuning' does not necessarily imply a higher signal amplitude and could be observable even under signal suppression. The authors should comment on this and clarify.

We appreciate your highlighting of these highly relevant alternative explanations. In response, we have revised a paragraph in the General Discussion on page 18 to explicitly outline our rationale for associating decreased tuning with proactive suppression. However, in doing so, we now also consider the alternative perspective that proactive suppression might actually require enhanced tuning towards the expected feature to implement suppression effectively.

It's important to note that both of these interpretations – decreased tuning as a sign of suppression and increased tuning as a preparatory mechanism for suppression – diverge significantly from the commonly held model (including our own initial assumptions) wherein weights at the to-be-suppressed location are simply downregulated.

Minor:(1) In the Word file I reviewed, there are minor formatting issues, such as missing spaces, which should be double-checked.

Thank you! We have now reviewed the text thoroughly and tried our best to avoid formatting issues.

(2) Would the authors predict that proactive mechanisms are not involved in other forms of attention learning involving distractor suppression, such as habituation?

Habituation is a form of non-associative learning where the response to a repetitive stimulus decreases over time. As such, we would not characterize these changes as “proactive”, as it only occurs following the (repeated) exposure to the stimulus.

(3) A clear description in the Methods section of how individual CTFs for each location were derived would help in understanding the procedure.

Thank you. We have now added several sentences on page 27 to clarify how individual CTFs in Figure 3 and distance CTFs in Figure 5 are calculated.

“The derived channel responses (8 channels × 8 location bins) were then used for the following analyses: (a) calculating individual Channel Tuning Functions (CTFs) based on each of the eight physical location bins (e.g., Figure 3C and 3D); (b) grouping responses according to the distance between each physical location and the high-probability distractor location to calculate distance CTFs (e.g., Figure 5); and (c) averaging across location bins to represent the general strength of spatial selectivity in tracking the memory cue, irrespective of its specific location (e.g., Figure 3A and 3B).”

(4) Why specifically 1024 resampling iterations?

Thank you for your question. The statistical analysis was conducted using *the permutation_cluster_1samp_test* function within the MNE package in Python. We have clarified this on page 25. The choice of 1024 permutations reflects the default setting of the function, which is generally considered sufficient for robust non-parametric statistical testing. This number provides a balance between computational efficiency and the precision of p-value estimation in the context of our analyses.

**Reviewer #3 (Public Review):**
Summary:In this experiment, the authors use a probe method along with time-frequency analyses to ascertain the attentional priority map prior to a visual search display in which one location is more likely to contain a salient distractor. The main finding is that neural responses to the probe indicate that the high probability location is attended, rather than suppressed, prior to the search display onset. The authors conclude that suppression of distractors at high-probability locations is a result of reactive, rather than proactive, suppression.Strengths:This was a creative approach to a difficult and important question about attention. The use of this "pinging" method to assess the attentional priority map has a lot of potential value for a number of questions related to attention and visual search. Here as well, the authors have used it to address a question about distractor suppression that has been the subject of competing theories for many years in the field. The paper is well-written, and the authors have done a good job placing their data in the larger context of recent findings in the field.Weaknesses:The link between the memory task and the search task could be explored in greater detail. For example, how might attentional priority maps change because of the need to hold a location in working memory? This might limit the generalizability of these findings. There could be more analysis of behavioral data to address this question. In addition, the authors could explore the role that intertrial repetition plays in the attentional priority map as these factors necessarily differ between conditions in the current design. Finally, the explanation of the CTF analyses in the results could be written more clearly for readers who are less familiar with this specific approach (which has not been used in this field much previously).

We appreciate the reviewer's valuable feedback and have made significant revisions to address the concerns raised. To clarify the connection between the memory and search tasks, we conducted additional analyses to explore the effects of spatial distance between the memory cue location and the high-probability distractor location on behavioral performance. We also investigated the potential influence of intertrial repetition effects on the observed results by removing trials with location repetitions. To enhance clarity, we revised the explanation of the CTF analyses in the Results section and improved figure annotations to ensure accessibility for readers unfamiliar with this approach. Collectively, these updates further discuss how the pattern of CTF slopes reflect the interplay between memory and search tasks while addressing key methodological and interpretative considerations.

**Recommendations for the authors:**

**Reviewer #1 (Recommendations For The Authors):**
Suggestions/Critiques (in no particular order)(1) The authors discuss the tripartite model (bottom-up, top-down, and selection history) but neglect recent and important discussions of why this trichotomy might be unnecessarily complicated (e.g., Anderson, 2024: Trichotomy revisited: A monolithic theory of attentional control). Simply put, one of the 3 pillars (i.e., selection history) likely does not fall into a unitary construct or "box"; instead, it likely contains many subcomponents (e.g., reward associations, stimulus-response habit learning, statistical learning, etc.). Since the focus of the current study is learned distractor suppression based on the statistical regularities of the distractor, the authors should comment on which aspects of selection history are relevant, perhaps by using this monolithic framework.

We appreciate the reviewer's insightful suggestion regarding theoretical frameworks of attentional control. While Anderson (2024) proposes a monolithic theory that challenges the traditional tripartite model, our study deliberately maintains a pragmatic approach. The main purpose of our experiment is empirically investigating the mechanisms of learned distractor suppression, rather than adjudicating between competing theoretical models.

We agree that selection history is not a unitary construct but comprises multiple subcomponents, including reward associations, stimulus-response habit learning, and statistical learning. In this context, our study specifically focuses on statistical learning as a key mechanism of distractor suppression. By explicitly acknowledging the multifaceted nature of selection history and referencing Anderson's monolithic perspective, we invite readers to consider the theoretical implications while maintaining our research's primary focus on empirical investigation. To this end, we have modified the manuscript to read (see page 3):

"The present study investigates the mechanisms underlying statistical learning, specifically learned distractor suppression, which represents one critical subcomponent of selection history. While theoretical models like the tripartite framework and the recent monolithic theory (Anderson, 2024) offer complementary perspectives on attentional control, our investigation focuses on empirically characterizing the statistical learning mechanisms underlying learned distractor suppression."

(2) The authors discuss previous demonstrations of location-based and feature-based learned distractor suppression. The authors admit that there have been a large number of studies but seem to mainly cite those that were conducted by the authors themselves (with the exception being Vatterott & Vecera, 2012). For example, there are other studies investigating location-based suppression (Feldmann-Wüstefeld et al., 2021; Sauter et al., 2021), feature-based suppression (Gaspelin & Luck, 2018a; Stilwell et al., 2022; Stilwell & Gaspelin, 2021; Vatterott et al., 2018), or both (Stilwell et al., 2019). The authors do not cite Gaspelin and colleagues at all in the manuscript, despite claiming that singleton-based suppression is not proactive.

We appreciate your pointing out the need for a more comprehensive citation of the literature on learned distractor suppression, particularly with respect to location-based and feature-based suppression. In response to your comment, we have now expanded the reference list on page 4 to include relevant studies that further support our discussion of both location-based and feature-based suppression mechanisms.

(3) The authors use the terms "proactive" and "reactive" suppression without taking into consideration the recent terminology paper, which one of the current authors, Theeuwes, helped to write (Liesefeld et al., 2024, see Figure 8). The terms proactive and reactive suppression need to be defined relative to a time point. The authors need to be careful in defining proactive suppression as prior to the first shift of attention, but after the stimuli appear and reactive suppression as after the first shift of attention and after the stimuli appear. Thus, the critical time point is the first shift of attention. Does suppression occur before or after the first shift of attention? The authors could alleviate this by using the term "stimulus-triggered suppression" to refer to "suppression that occurs after the distractor appears and before it captures attention" (Liesefeld et al., 2024).

Thank you for pointing out that this was insufficiently clear in the previous version. In the revised version we specifically refer to the recent terminology paper on page 5 to make clear that suppression could theoretically occur at three distinct moments in time, and that the present paper was designed to dissociate between suppression before or after the first shift of attention.

(4) Could the authors justify why the circle stimulus (2° in diameter) was smaller than the diamonds (2.3° x 2.3°)? Are the stimuli equated for the area? Or, for width and height? Doesn't this create a size singleton target on half of all trials (whenever the target is a circle) in addition to the lone circle being a shape singleton? Along these lines, could the authors justify why the colors were used and not equiluminant? This version of red is much brighter than this version of green if assessed by a spectrophotometer. Thus, there are sensory imbalances between the colors. Further, the grey used as the ping is likely not equiluminant to both colors. Thus, the grey "ping" is likely dimmer for red items but brighter for green items. Is this a fair "ping"?

Thank you for raising these important points. We chose, as is customary in this experimental paradigm (e.g., Huang et al., 2023; Duncan et al., 2023), to make the diamond slightly larger (2.3° x 2.3°) than the circle (2° in diameter) to ensure a better visual match in overall size appearance. If the circle and diamond stimuli were equated strictly in terms of size (both at 2°), the diamond would appear visually smaller due to the differences in geometric shape. By adjusting the dimensions slightly, we aimed to minimize any unintentional differences in perceptual salience.

As for the colors used in the experiment, the reviewer is right that there might be sensory imbalances between the red and green stimuli, with red appearing brighter than green based on measurements such as spectrophotometry. To ensure that any effects couldn’t be explained by sensory imbalance in the displays, we randomized target and distractor colors across trials, meaning that roughly half the trials had a red distractor and half had a green distractor. This randomization should have mitigated any systematic biases caused by color differences.

We appreciate your feedback and have clarified these points in method section in the revised manuscript on page 22:

"Please note that although the colors were not equiluminant, the target and distractor colors were randomized across trials such that roughly half the trials had a red distractor, and half had a green distractor. This randomization process should help mitigate any systematic biases this may cause."

(5) For the eye movement artifact rejection, the authors use a relatively liberal rejection routine (i.e., allowing for eye movements up to 1.2° visual angle and a threshold of 15 μV). Given that every 3.2 μV deviation in HEOG corresponds to ~ ± 0.1° of visual angle (Lins, et al., 1993), the current oculomotor rejection allows for eye movements between 0.5° and 1.2° visual angle to remain which might allow for microsaccades (e.g., Poletti, 2023) to contaminate the EEG signal (e.g., Woodman & Luck, 2003).

The reviewer correctly points out that our eye rejection procedure, which is the same as in our previous work (e.g., Duncan et al., 2023), still allows for small, but systematic biases in eye position towards the remembered location and potentially towards or away from the high probability distractor location. While we cannot indefinitely exclude this possibility, we believe this is unlikely for the following reasons. First, although there is a link between microsaccades and covert attention, it has been demonstrated that subtle biases in eye position cannot explain the link between alpha activity and the content of spatial WM (Foster et al., 2016, 2017). Specifically, Foster et al. (2017) found no evidence for a gaze-position-related CTF, while an analysis on that same data yielded clear target related CTFs. Similarly, within the present data set there was no evidence that the observed revival induced by the ping display could be attributed to systematic changes in gaze position, as a multivariate cross-session decoding analysis with x,y positions from the tracker did not yield reliable above-chance decoding of the location in memory.

**Author response image 1. sa4fig1:** 

(6) The authors claim that "If the statistically learned suppression was spatial-based and feature-blind, one would also expect impaired target processing at the high-probability location." (p. 7, lines 194-195). Why is it important that suppression is feature-blind here? Further, is this a fair test of whether suppression is feature-blind? What about inter-trial priming of the previous trial? If the previous trial's singleton color repeated RTs might be faster than if it switched. In other words, the more catastrophic the interference (the target shape, target color, distractor shape, distractor color) change between trials, the more RTs might slow (compared with consistencies between trials, such that the target and distractor shapes repeat and the target and distractor colors repeat). Lastly, given the variability across both the shape and color dimensions, the claim that this type of suppression is feature-blind might be an artifact of the design promoting location-based instead of feature-based suppression.

Thank you for raising this point. In the past we have used the finding that learned suppression was not specific to distractors, but also generalized to targets to argue in favor of proactive (or stimulus triggered) suppression. However, we agree that given the current experimental parameters it may be an oversimplification to conclude that the effect was feature-blind based on the impaired target processing as observed here. As this argument is also not relevant to our main findings, we have removed this interpretation and simply report that the effect was observed for both distractor and targets. Nevertheless, we would like to point out that while inter-trial priming could influence reaction times, the features of both target and distractors (shape and color) were randomly assigned on each trial. This should mitigate consistent feature repetitions effects. Additionally, previous research has demonstrated that suppression effects persist even when immediate feature repetitions are controlled for or statistically accounted for (e.g., Wang & Theeuwes 2018 JEP:HPP; Huang et al., 2021 PB&R).

(7) The authors should temper claims such as "suppression occurs only following attentional enhancement, indicating a reactive suppression mechanism rather than proactive suppression." (p. 15, lines 353-353). Perhaps this claim may be true in the current context, but this claim is too generalized and not supported, at least yet. Further, "Within the realm of learned distractor suppression, an ongoing debate centers around the question of whether, and precisely when, visual distractors can be proactively suppressed. As noted, the idea that learned spatial distractor suppression is applied proactively is largely based on the finding that the behavioral benefit observed when distractors appear with a higher probability at a given location is accompanied by a probe detection cost (measured via dot offset detection) at the high probability distractor location (Huang et al., 2022, 2023; Huang, Vilotijević, et al., 2021)." (p. 15, lines 355-361). Again, the authors should either cite more of the opposing side of the debate (e.g., the signal suppression hypothesis, Gaspelin & Luck, 2019 or Luck et al., 2021 and the many lines of converging evidence of proactive suppression) or temper the claims.

Thank you for your constructive feedback regarding our statements on suppression mechanisms. We acknowledge that our original claim was intended to reflect our specific findings within the context of this study and was not meant to generalize across all research in the field. To prevent any misunderstanding, we have tempered our claims to avoid overgeneralization by clarifying that our findings suggest a tendency toward reactive suppression within the specific experimental conditions we investigated (see page 17).

Furthermore, learned distractor suppression is multifaceted, encompassing both feature-based suppression (as proposed by the signal suppression hypothesis) and spatial-based suppression (as examined in the current study). The signal suppression hypothesis provides proactive evidence related to the suppression of specific feature values (Gaspelin et al., 2019; Gaspelin & Luck, 2018b; Stilwell et al., 2019). We have incorporated references to these studies to offer a more comprehensive perspective on the ongoing debate at a broader level (see page 17).

(8) "These studies however, mainly failed to find evidence in support of active preparatory inhibition (van Moorselaar et al., 2020, 2021; van Moorselaar & Slagter, 2019), with only one study observing increased preparatory alpha contralateral to the high probability distractor location (Wang et al., 2019)." (p. 15, lines 367-370). This is an odd phrasing to say "many studies" have shown one pattern (citing 3 studies) and "only" one showing the opposite, especially given these were all from the current authors' labs.

Agreed. We have rewritten this text on page 17.

“These studies however, failed to find evidence in support of active preparatory inhibition as indexed via increased alpha power contralateral to the high probability distractor location (van Moorselaar et al., 2020, 2021; van Moorselaar & Slagter, 2019; but see Wang et al., 2019).”

(9) Could the authors comment on why total power was significantly above baseline immediately (without clearer timing marks, ~10-50 ms) after the onset of the cue (Figure 3)? Is this an artifact of smearing? Further, it appears that there is significant activity (as strong as the evoked power of interest) in the baseline period of the evoked power when the memory item is presented on the vertical midline in the upper visual field (this is also true, albeit weaker, for the memory cue item presented on the horizontal midline to the right). This concern again appears in Figure 4 where the Alpha CTF slope was significantly below or above the baseline prior to the onset of the memory cue. Evoked Alpha was already significantly higher than baseline in the baseline period. In Figure 5, evoked power is already higher and different for the hpl than the lpls even at the memory cue (and before the memory cue onsets). There are often periods of differential overlap during the baseline period, or significant activity in the baseline period or at the onset of the critical, time-locked stimulus array. The authors should explain why this might be (e.g., smearing).

Thank you for pointing this out. As suggested by the reviewer, this ‘unexpected’ pre-stimulus decoding is indeed the result of temporal smearing induced by our 5th order Butterworth filter. The immediate onset of reliable tuning (sometimes even before stimulus onset) is then also a typical aspect of studies that track tuning profiles across time in the lower frequency bands such as alpha (van Moorselaar & Slagter 2019; van Moorselaar et al., 2020; Foster et al., 2016).

Indeed, visual inspection also suggests that evoked activity tracked items at the top of the screen, an effect that is unlikely to result from temporal smearing as it is temporally interrupted around display onset. However, it is important to note that CTFs by location are based on far fewer trials, making them inherently noisier. The by-location plots primarily serve to show that the observed pattern is generally consistent across locations. In any case, given that the high probability distractor location was counterbalanced across participants it did not systematically influence our results.

(10) Given that EEG was measured, perhaps the authors could show data to connect with the extant literature. For example, by showing the ERP N2pc and PD components. A strong prediction here is that there should be an N2pc component followed by a PD component if there is the first selection of the singleton before it is suppressed.

Thank you for your great suggestion regarding the analysis of ERP components such as N2pc and Pd. To reliably assess lateralized ERP components like N2pc or Pd the high probability location must be restricted to static lateralized positions (e.g., on the horizontal midline such as Wang et al., 2019). In contrast, our study was designed to utilize an inverted encoding model to investigate the mechanisms underlying spatial suppression. To avoid bias in training the spatial model toward specific spatial locations (see also the previous comment), we counterbalanced the high-probability location across participants, ensuring an equal distribution of high-probability locations within the sample. Given this counterbalanced design, it was not feasible to reliably assess these components within the scope of the current study. Yet, we agreed with the reviewer that it would be of theoretical interest to examine Pd and N2pc evoked by the search display, particularly in this scenario where suppression has been triggered prior to search onset.

(11) Figure 2 (behavioral results) is difficult to see (especially the light grey and white bars). A simple fix might be to outline all the bars in black.

Thank you! We have incorporated your suggestion by outlining all the bars on page 10.

**Reviewer #3 (Recommendations For The Authors):**(1) I'm wondering about the link between the memory task and the search task. I think the interpretation of the data should include more discussion of the fact that much of the search literature doesn't involve simultaneously holding an unrelated location in memory. How might that change the results?For example - what happens behaviorally on the subset of trials in which the location to be held in memory is near the high probability distractor location? All the behavioral data is more or less compartmentalized, but I think some behavioral analysis of this and related questions might be quite useful. I know there are comparisons of behavior in single vs. dual-task cases (for the memory task at least), but I think the analyses could go deeper.

Thank you for your great suggestion. To investigate the potential interactions between the spatial memory task and the visual search task, we conducted additional analyses on the behavioral data. First, we examined whether memory recall was influenced by the spatial distance (dist0 to dist4) between the memory cue location and the high-probability distractor location. As shown in the figure below, memory recall is not systematically biased either toward or away from the high-probability distractor location (p = .562, ηp^2^ = .011).

We also assessed how the memory task might affect search performance. Specifically, we plotted reaction times as a function of the spatial overlap between the memory cue location and any of the search items, separating trials by distractor-present (match-target, match-distractor, match-neutral) and distractor-absent (match-target, match-neutral) conditions. Although visually the result pattern seems to suggest that search performance was facilitated when the memory cue spatially overlapped with the target and interfered with when it overlapped with the distractor, this pattern did not reach statistical significance (distractor-present: p = .249, ηp^2^ = .002; distractor-absent: p = .335, ηp^2^ = .002). We have now included these analyses in our supplemental material.

Beyond additional data analyses, there are also theoretical questions to be asked. For example, one could argue that in order to maintain a location near or at the high probability distractor location in working memory, the priority map would have to shift substantially. This doesn't necessarily mean that proactive suppression always occurs in search when there is a high probability location. Instead, one could argue that when you need to maintain a high probability location in memory but also know that this location might contain a distractor, the representation necessarily looks quite different than if there were no memory tasks. Maybe there are reasons against this kind of interpretation but more discussion could be devoted to it in the manuscript. I guess another way to think of this question is - how much is the ping showing us about attentional priority for search vs. attentional priority for memory, or is it simply a combination of those things, and if so, how might that change if we could ping the attentional priority map without a simultaneous memory task?

Thank you for this valuable suggestion. The aim of our study was to explore how the CTFs elicited by the memory cue were influenced by the search task. We employed a simultaneous memory task because directly measuring CTFs in relation to the search task was not feasible, as the HPL typically does not vary within individual participants. Consequently, CTFs locked to placeholder onsets could reflect arbitrary differences between (subgroups of) participants rather than true differences in the HPL. To address this, we combined the search task with a VWM task, leveraging the fact that location-specific CTFs can reliably be elicited by a memory cue and that the location of this cue relative to the HPL can be systematically varied within participants (Foster et al., 2016, 2017; van Moorselaar et al., 2018). This approach allowed us to examine the CTFs elicited by the memory cue and how these were modulated by their distance from the HPL.

While it is theoretically possible that the observed changes resulted from alterations in how the memory cue was maintained in memory only, this explanation seems unlikely, for memory performance (recall) did not vary as a function of the cue's distance from the HPL, suggesting that the distance-related changes in the CTFs are reflections of both tasks. Moreover, distractor learning typically occurs without awareness (Gao & Theeuwes 2022; Wang & Theeuwes 2018). It is difficult to understand how such unconscious processes could lead to anticipations in the memory task and subsequently modulate the representation of the consciously remembered memory cue only. We therefore believe that if we would have pinged the attentional priority map without a simultaneous memory task, the results would have been similar to those obtained in the present experiment, indicating stronger tuning at the HPL. Yet, this work still needs to be done.

To address this comment, we have added a paragraph on p. 18:

“However, two alternative explanations warrant consideration. First, one could argue that observed modulations in the revived CTFs do not provide insight into the mechanisms underlying distractor suppression but instead reflect changes in the memory representation itself, potentially triggered by the anticipation of the HPL in the search task. According to this view, the changes in the revived CTFs would be unrelated to how search performance (in particular distractor suppression) was achieved. While this is theoretically possible, we believe it to be unlikely. Memory performance (recall) did not vary as a function of the cue's distance from the HPL, whereas the revived CTFs did, indicating that these changes likely reflect contributions from both tasks. Additionally, distractor learning typically occurs without conscious awareness (Gao & Theeuwes 2022; Wang & Theeuwes 2018). It is difficult to conceive how such unconscious processes could produce anticipatory effects in the memory task and selectively modulate the representation of the consciously remembered memory cue. Second, the apparent lack of suppression and the presence of a pronounced tuning at the high-probability distractor location could actually reflect a proactive mechanism that manifests in a way that seems reactive due to the dual-task nature of our experiment.”

(2) When the distractor appears at a particular location with a high probability it necessarily means that intertrial effects differ between high and low probability distractor locations. Consecutive trials with a distractor at the same location are far more frequent in the high probability condition. You may not have enough power to look at this, and I know this group has analyzed this behaviorally in the past, but I do wonder how much that influences the EEG data reported here. Are CTFs also sensitive to distractors/targets from the most recent trial? And does that contribute to the overall patterns observed here?

Thank you for your thoughtful comment. Indeed, Statistical distractor learning studies naturally involve a higher proportion of intertrial effects for high-probability distractors compared to low-probability ones. Previous research, including the present study, has demonstrated that while distractor location improves performance—shown by faster response times (t(23) = 6.32, p < .001, d = 0.33) and increased accuracy (t(23) = 4.21, p < .001, d = 0.86)—intertrial effects alone cannot fully account for the learned suppression effects induced by spatial distractor imbalances. This analysis in now reflected in the revised manuscript on page 9.

However, as noted by the reviewer, this leaves uncertain to what extent the neural indices of statistical learning, in this case the modulation of channel tuning functions, capture the effects of interest beyond the contributions of intertrial priming. To address this issue, one possible approach is to rerun the CTF analysis after excluding trials with location repetitions. Since the distractor location is unknown to participants at the time the CTF is revived by the placeholder, we removed trials where the memory cue location repeated the distractor location from the preceding trial, rather than trials with distractor location repetitions between consecutive trials. Our analyses indicate that after trials removal (~ 9% of overall trials), the spatial gradient pattern in the CTF slopes remains similar. However, the cluster-based permutation analysis fails to reveal any significant findings, and a one-sample t-test on the slopes averaged within the 100 ms time window of interest yields a p-value of 0.106. While this could suggest that the current pattern is influenced by distractor-cue repetition, it is more likely that the trial removal resulted in an underpowered analysis. To investigate this, we randomly removed an equivalent number of trials (9%), which similarly resulted in insignificant findings, although the overall result pattern remained comparable (p = 0.066 for the one-sample t-test on the slopes average within the interested time window of 100 ms).

**Author response image 2. sa4fig2:** 

Also, in our previous pinging study we observed that, despite the trial imbalance, decoding was approximately equal between high probability trailing (i.e., location intertrial priming) and non-trailing trials, suggesting that the ping is able to retrieve the priority landscape that build up across longer timescales.

(3) Maybe there is too much noise in the data for this, but one could look at individual differences in the magnitude of the high probability distractor suppression and the magnitude of the alpha CTF slope. If there were a correlation here it would bolster the argument about the relationship between priority to the distractor location and subsequent behavior reduction of interference from that distractor.

Thank you for this valuable suggestion. We investigated whether there was a correlation between the average gradient slope during the time window in which the placeholder revived the memory representation and the average distance slope in reaction times for the learned suppression effect. This correlation was not significant (r = .236, p = 0.267), which is perhaps expected given the potential noise levels, as noted by the reviewer. Furthermore, while the learned suppression effect is robust at the group level, its predictive value for individual-level performance has been shown to be limited (Ivanov et al., 2024; Hedge et al., 2018). Consequently, we chose not to include this analysis in the manuscript (see also our response to comment 2 by reviewer 2).

(4) The results sections are a bit dense in places, especially starting at the bottom of page 11. For readers who are familiar with the general questions being asked but less so with the particular time-frequency analyses and CTF approaches being used (like myself), I think a bit more time could be spent setting up these analyses within the results section to make extra clear what's going on.

Thank you for your feedback regarding the clarity of our Results section. We have revised this section to make it more understandable and easier to follow, especially for readers who may be less familiar with the specific time-frequency analyses and modeling approaches used in our study. Specifically, we have provided additional interpretations alongside the reported results from page 10 to page 13 to aid comprehension and ensure that the methodology and findings are accessible to a broader audience. Additionally, we have revised the figure notes to further enhance clarity and understanding.

Other comments:Abstract: "a neutral placeholder display was presented to probe how hidden priority map is reconfigured..." i think the word "the" is missing before "priority map"

Thank you. We have added the word “the” before “hidden priority map”.

p. 4, Müller's group also has a number of papers that demonstrate how learned distractor regularities impact search (From the ~2008-2012 range, probably others as well), it might be worth citing a few here.

Thank you for your suggestion. In the revised manuscript, we have added citations to several key papers from Muller’s group on page 4 as well as other research groups.

p.5 - Chang et al. (2023) seems highly relevant to the current study (and consistent with its results) - depending on word limits, it might make sense to expand the description of this in the introduction to make clear how the present study builds upon it

Thank you! We have expanded the discussion of Chang et al. (2023) on page 5 to provide more detailed elaboration of their study and its relevance to our work.

p. 7 - maybe not for the current study, but I do wonder whether the distortion of spatial memory by the presence of the search task occurs only when there is a relevant regularity in the search task. In other words, if the additional singleton task had completely unpredictable target and distractor locations, would there be memory distortions? Possibly for the current dataset, the authors could explore whether the behavioral distortion is systematically towards or away from the high probability distractor location.

Thank you for your insightful suggestion. Following your recommendation, we conducted an additional analysis to examine memory recall as a function of the distance between the memory cue location and the high-probability distractor location. Figure S1A illustrates the results, depicting memory recall deviation across various distances (dist0 to dist4) from the high-probability distractor location.

Our statistical analysis indicates that memory recall is not systematically biased either towards or away from the high-probability distractor location (*p* = .562, η_p_^2^ = .011). This finding suggests that spatial memory recall remains relatively stable and is not heavily influenced by the presence of regularities in the distractor locations.

p. 7 - in addition to stats it would be helpful to report descriptive statistics for the high probability vs. other distractor location comparisons

Thank you! We have added descriptive statistics on page 8 and page 9.

p. 19, "64%" repeated unnecessarily - also, shouldn't it be 65% if it's 5% at each of the other seven locations?

Thank you. This is now corrected in the revised manuscript.

p. 20 "This process continued until participants demonstrated a thorough understanding of the assigned tasks" Were there objective criteria to measure this?

Thank you for pointing out this issue. To clarify, objective criteria were indeed used to assess participants’ readiness to proceed. Specifically:

For the training phase practice trials, participants were required to achieve an average memory recall deviation of less than 13°.

For the test phase practice trials, participants needed to demonstrate a minimum of 65% accuracy in the search task. In addition, participants were asked to verbally confirm their understanding of the task goals with the experimenter before proceeding.

We have revised the manuscript to clearly indicate these criteria on p. 23.

p. 21 "P-values were Greenhouse-Geiser corrected in case where the..." I think "case" should be "cases"

Thank you. We have corrected this in the revised manuscript.